# High-resolution diapycnal mixing map of the Alboran Sea thermocline from seismic reflection images

Jhon F. Mojica[1-2], Valentí Sallarès[2], Berta Biescas[2-3]

[1] Center for global Sea Level Change CSLC – NYUAD, Abu Dhabi UAE.
[2] Institute of Marine Sciences, ICM-CSIC, Barcelona, Spain
[3] Consiglio Nazionale delle Ricerche CNR-ISMAR, Bologna, Italy

*Correspondence to:* Jhon F. Mojica (jhon.mojica@nyu.edu)

**Abstract.** The Alboran Sea is a dynamically active region where the salty and warm Mediterranean water first encounters the incoming milder and cooler Atlantic water. The interaction between these two water masses originates a set of sub-mesoscale structures and a complex sequence of
processes that entail mixing close to the thermocline. Here we present a high-resolution map of the diapycnal diffusivity around the thermocline depth obtained using acoustic data recorded with a high-resolution multichannel seismic system. The map reveals a patchy thermocline, with spots of strong diapycnal mixing juxtaposed with areas of weaker mixing. The patch size is of a few kms in the horizontal scale and of 10-15 m in the vertical one. The comparison of the obtained maps
with the original acoustic images shows that mixing tends to concentrate in areas where internal waves, which are ubiquitous in the surveyed area, become unstable and shear instabilities develop, enhancing energy transfer towards the turbulent regime. These results are also compared with others obtained using more conventional oceanographic probes. The values estimated based on the seismic data are within the ranges of values obtained from oceanographic data analysis, and they
are also consistent with reference theoretical values. Overall, our results demonstrate that high-resolution seismic systems allow to remotely quantifying mixing at the thermocline depth with unprecedented resolution.

KEYWORDS: Thermocline mixing, Seismic Oceanography, Diapycnal mixing map.

**1. INTRODUCTION**

Diapycnal diffusivity ($k_\rho$) around the thermocline plays a major role to control the strength and pattern of the ocean circulation, because it determines heat and salt heterogeneity at different spatial scales. This process usually occurs in a vertically stratified regime, affecting adjacent layers
with the same density but different temperature and salinity (Stewart, 2008). In terms of processes, mixing in the ocean can be separated in two categories. One corresponds to internal wave (IW) breaking, which produces turbulent motion and changes the density stratification, while the second concerns the development of high frequency dynamic instabilities that are formed due to shear (Gregg, 1987; D'Asaro and Lien, 2000). As the spatial scale decreases, mixing leads to an
unbalanced pressure field that eventually results in a collapse and dispersion of mixing waters through isopycnals (Thorpe, 2005). The value of $k_\rho$ depends on the buoyancy frequency ($N$) and the dissipation rate ($\varepsilon$) as indicated by the so-called Osborn (1980) relationship:

$$k_\rho = \Gamma \varepsilon / N^2 \tag{1}$$


This value, where $\Gamma = 0.2$ is the empirically defined mixing efficiency (Osborn and Cox, 1972), corresponds to the mixing between isopycnal layers in the thermocline. The global mean $k_\rho$ value is of the order of $10^{-4}\,m^2s^{-1}$ (Munk and Wunsch, 1998), which corresponds to the value required to keep overturning in the thermocline. It has been shown that if $k_\rho < 10^{-5}\,m^2s^{-1}$, the energy is not
enough to generate mixing (Gregg, 1989).

In a conservative flow, $\varepsilon$ might present small variation due to dissipated heat through turbulent motions, but in the presence of strong shear $\varepsilon$ tends to increase (Thorpe, 2005), reaching a maximum value close to the Kolmogorov scale (Gargett and Holloway, 1984). Good knowledge
of its behavior provides important clues on available energy and its transfer between spatial scales.

The loss rate of kinetic energy in the turbulent motion is commonly expressed as:

$$\varepsilon = \left(\frac{\nu}{2}\right)\langle S_{ij}S_{ij}\rangle \tag{2}$$


$$S_{ij} = \left(\frac{\partial u_i}{\partial x_j} + \frac{\partial u_j}{\partial x_i}\right) \tag{3}$$

Where $\nu = 1.064x10^{-6}\,m^2s^{-1}$ is the kinematic viscosity and the tensor $S_{ij}$ is a function of the velocity components in the three orthogonal directions (Thorpe, 2005). Conventional in-situ techniques as
Vertical Microstructure turbulence Profiler (VMP) or microriders provide the most accurate measures of $k_\rho$, but in just one dimension. In general, although measures are accurate in the vertical dimension, sampling in the horizontal direction is much poorer, particularly in the $\sim 10^3\text{-}10^1\,m$ range (Klymak and Moum, 2007 a, b). Since this is the range of scales at which the transition between isotropic internal wave and anisotropic turbulent motions (i.e. mixing) occurs, the
observational evidence of mixing patterns and the understanding of the underlying physical mechanisms are rather limited so far. Overall, direct measures and observations are too few to create a global mixing map with the required resolution to feed the models with proper values of dissipation rates (Smyth et al., 2011). In turn this makes it difficult to integrate mixing into large-scale models of ocean dynamics. Mixing effects are simulated instead through the incorporation
of eddy diffusivity coefficients, which are tuned *ad hoc* to match the large-scale distribution of ocean observables. While this approach allows to properly reproduce regional spatial-temporal patterns, it severely hampers the long-term predictive capability of ocean dynamics and, in turn, that of climate models. Thus improving our knowledge on the short-term and small-scale mixing mechanisms, and integrating them into large-scale models, remain as an outstanding challenge.

To overcome this issue, remote sensing techniques have recently started to be used (e.g. Gibson et al., 2007). One of these alternative techniques is multichannel seismics (MCS), an acoustic method providing quasi-synoptic images of the thermohaline boundaries in the ocean interior to full ocean depth, with a lateral resolution of O($10^1\,m$) (Holbrook et al., 2003). Several recent works have
demonstrated that it is actually possible to map $k_\rho$ using measures of the horizontal wavenumber ($k_x$) spectra of the vertical displacements of thermohaline boundaries imaged with MCS acquisition systems (Sheen et al., 2009; Holbrook et al., 2013; Fortin et al., 2016). However, these studies use conventional, relatively low-resolution systems with source energy concentrating below $\sim 50\,Hz$. In addition, due to the long wavelength source wavelet, conventional MCS systems are not well
suited to image the shallowest ocean layers (i.e. $< 200\,m$), but deeper water levels ($\geq 400\,m$ depth).

At these depth levels, changes in the internal structure are usually less marked than at shallower levels, especially around the thermocline. In a recent work, it has been shown that portable, high resolution MCS (HR-MCS) systems, which use a smaller energy, but higher-frequency source (>150 Hz), allow imaging the thermohaline structure as shallow as *~30 m* with a lateral resolution of *12-15 m* and a vertical one of *1-2 m* (Sallares et al., 2016). This resolution is three- to four-fold better than that of conventional MCS systems that have been used to image deeper ocean levels. Therefore, it has the potential to image sub-mesoscale structures and processes that affect the thermocline at scales of kilometers to tens of meters, allowing coverage of the existing observational gap. Despite its potential, HR-MCS systems have never been used to date to quantify diapycnal mixing at the thermocline depth.

Here we use the above-mentioned method of extracting $k_\rho(x,z)$ maps from MCS images, but applied for the first time to HR-MCS data acquired in the Alboran Sea (Westernmost Mediterranean). The method to calculate diapycnal mixing maps from the horizontal wavenumber spectra of vertical reflector displacements is based on that proposed by Sheen et al. (2009) and Holbrook et al. (2013). The result is a high-resolution mixing map of the ocean at the thermocline depth (*30-110 m*) along a *35 km*-long transect (Fig. 1a). This method could be applied in other regions where the shallow water levels are sufficiently stratified to allow recording the energy reflected at variations of acoustic impedance (density x sound speed contrasts between neighboring water layers).

The rest of the manuscript is structured as follows: in section 2 we present the hydrographic context and the observations; then we describe the acquisition system and the method applied to estimate $k_\rho$ from the seismic data. The results are described in section 3, whereas the discussion about the imaged structures and their likely causes is presented in section 4. Finally, section 5 summarizes the main conclusions.

## 2. DATA AND METHODOLOGY

The Alboran Sea is characterized by the continuous exchange between Mediterranean Water (MW) and Atlantic Water (AW) through the Strait of Gibraltar. This exchange concentrates near the surface (between *~30 m* and *~200 m*); where the shallow, moderately salty and cold incoming AW (*< 50 m*) interacts with the deeper, warmer, saltier and more stable outgoing MW, producing another water mass known as Modified Atlantic Water (MAW). In this framework, internal waves, strong horizontal shear instability, and prominent thermohaline stratification are generated. These particular features reflect the complex dynamic setting of the area, with kinetic energy being transferred between isopycnals from large to small scales, leading eventually to overturning, isotropic turbulence and irreversible mixing.

The data set used in this work, which includes collocated seismic and oceanographic measurements, was collected on board the Spanish R/V Hesperides in the framework of the IMPULS-2006 experiment. Here, we concentrate our analysis on one of the seismic profiles, acquired using a portable HR-MCS system (IMPULS-3 profile). The acquisition started on May 16[th] at 23:43 and finished on May 17[th] at 04:00. In total, some 4 hours to record a 38 km-long profile. The acquisition system consisted of a 4.75 liters source with a peak frequency at 150-190 Hz. As mentioned above, the corresponding size of the Fresnel zone, a proxy of the horizontal

resolution, is *12-15 m* depending on the target depth. The streamer was 300 m-long and had 48 channels, with a group spacing of 6.25 m. The shot interval was 15 m, giving a Common Mid-Point (CMP) gather fold of 6. The location of the different data is displayed in Fig. 1a.


This profile was first processed and then used to estimate the average $k_x$ energy spectra of vertical displacements of the imaged reflectors (i.e. the acoustic images of thermohaline boundaries). A total of 68 reflectors rather homogeneously distributed throughout the surveyed area, with lengths of *1.5-21 km,* and a signal-to-noise ratio higher than *8* within the frequency range of *40–240 Hz*

(Sallares et al., 2016), were tracked and used for the analysis (Fig. 2). Vertical profiles of temperature and pressure were recorded simultaneously with the seismic acquisition using 4 expendable bathy-thermographs (XBTs); whereas the salinity and buoyancy profiles were derived from an expendable conductivity-temperature-depth (XCTD) probe dropped three days after the seismic acquisition. Water current profiles recorded with an Acoustic Doppler current profiler

(ADCP) in the same season as the seismic experiment, but 4 years later (see location in Fig. 1a), have also been used.

The IMPULS-3 profile shown in Fig. 2 reveals a number of laterally coherent seismic reflectors that are assumed to follow isopycnals. Biescas et al. (2014) showed that this assumption is valid

in regions free of salinity-temperature compensating intrusions, which is a reasonable approximation for the Alboran Sea. The analysis of the obtained $k_x$ spectra have allowed identifying three subranges that control dynamics around the thermocline depth (Sallares et al., 2016). At scales larger than the horizontal buoyancy wavelength ($l_N \approx 90\ m$), motions are dominated by the internal wave (IW) field (internal wave-field subrange). Then the spectra rolls off reflecting

the presence of shear instabilities of probably the Kelvin-Helmholtz (KH) type, which appear to collapse at a scale of *~30 m* (transitional, or instability-dominated subrange), giving rise to turbulence at even smaller scales (turbulent subrange) (Fig. S3). A more detailed description of these ranges and their scales of influence is presented in Sallares et al. (2016). In the present work, we use energy levels at the turbulent subrange, obtained from the $k_x$ spectral analysis of the tracked

reflectors within small analyzing windows, to estimate the lateral and vertical variations of $\varepsilon$ and $k_\rho$ along the whole profile.

Since our dataset does not include direct measurements of turbulence, we use the XCTD and ADCP data to estimate a vertical profile of $k_\rho$ based on Gregg's (1989) model, hereafter referred

to as Gregg89. This model assumes that energy dissipation in the thermocline takes place through IW energy transfer by wave-wave interaction. As the relative local change in buoyancy frequency, which relates the level of stretching and squeezing of isopycnals, is small in the surveyed area (≈1), we assume that the assumptions and approximations of the model are valid, so it is not necessary to consider alternative ones such as the one presented in Waterman et al. (2013). The

Gregg89 model is commonly applied in the mid-latitude thermocline, linking shear current at different depths. The simplest way to obtain average dissipation rates over large space and time scales is through:

$$\varepsilon = 7x10^{-10}\ N^2/N_0^2 < S_{10}^4/S_{GM}^4 > \tag{4}$$


$$S_{10}^4 = 4.22[(\Delta U/\Delta z)^2 + (\Delta V/\Delta z)^2]^2 \tag{5}$$

$$S_{GM}^4 = 2[(3\pi/2)j_x E_{GM} b N_0^2 k_x^c (N/N_0)^2]^2 \tag{6}$$

Where $N_0 = 5.2 \times 10^{-3}$ $s^{-1}$ is the reference buoyancy frequency, $S_{10}$ is the shear variance calculated from the meridional ($V$) and zonal ($U$) velocity variations as a function of depth ($z$), $S_{GM}$ is the variance for the Garret-Munk model (Gregg, 1989), $j_x$ is a mode number, $E_{GM}$ is the Garrett-Munk energy density, $b$ is the scale depth of the thermocline, $c$ is the spectrum slope, and $k_x$ is the horizontal wavenumber.

Alternatively, the model proposed by Batchelor (1959); hereafter referred to as Batchelor59, estimates $k_\rho$ as a function of the energy transfer from large to small scales in the turbulent regime. This model assumes that the energy exchange from mechanical to caloric due to $N$ and $\varepsilon$ can be approximated as:

$$\varphi_\varsigma^T = \left(\frac{4\pi\Gamma}{N^2}\right) C_T \varepsilon_T^{2/3} (2\pi k_x)^{-5/3} \tag{7}$$

Where $\varphi_\varsigma$ is the energy spectrum of the isopycnals vertical displacement measured in the turbulent subrange; and $C_T$ is a proportionality constant (Sreenivasan, 1996). We apply this model to estimate the mixing rates over the seismic profiles, applying a method proposed and described in previous works (e.g. Sheen et al., 2009; Holbrook et al., 2013). The main steps of the approach and the specifics of our work are described below.

 (1) Selecting a local window larger than the resolution of the data, but smaller that the entire seismic transect. The point is selecting the smallest possible window that allows proper calculation of the reflector displacement spectra. We tested different window sizes and we found that the smallest ones that allow producing robust results are *1200 m* wide x *15 m* high. Results with larger windows are comparable in terms of amplitude and shape of the imaged features, but structures and boundaries are better defined with this window size (Fig. S1). Smaller windows contain too few reflectors and produce abundant artefacts. Longer tracks are cut into shorter segments to fit inside the window. As explained in Sallares et al. (2016) and shown in Fig. S2, this does not affect the spectral values at the spatial scale range analyzed.

(2) Computing the energy level of the displacement spectra in the turbulent subrange for all the reflectors inside the window, and calculate the average spectrum. The spectral subranges observed in the combined spectrum of the 68 reflectors (Fig. S3), which are also observed in most individual windows and reflectors, are used as a reference to select the scale range to compute the spectral amplitudes. In the case of the turbulent subrange, it is *13-30* m. The mean number of reflectors fitting inside the *1200 m* wide x *15 m* high windows is three, ranging from two to four depending on the imaged area.

(3) Applying the Batchelor59 model (Eq. 7) to estimate $\varepsilon$, using the turbulent energy level (i.e. the average $\varphi_\varsigma$ between *13-30* m) computed within the window, with $\Gamma=0.2$, $C_T=0.3$, and $N$ is calculated according to depth. Finally, we apply Osborn80 relationship (Eq. 1) using the $\varepsilon(x, z)$ values obtained above to compute $k_\rho(x, z)$.

(4) As only few tracked reflectors are included inside each window, variances can affect the calculated diffusivities. To mitigate this effect, we slide the window in small steps (30 m in

horizontal and 3 m in vertical direction), assigning the average value of the spectral amplitude to each local window. The fact that we incorporate few new data at each step, produces a smoothly varying map with a resolution that is similar to the window size (*~1000x10 m*), instead of the one with sharp boundaries that is obtained without using overlapping windows (Fig. S1).

In summary, on one hand we apply the Gregg89 model to obtain a vertical $k_\rho(z)$ profile using the XCTD and ADCP data, and on the other hand we apply Batchelor59 to obtain a $k_\rho(x,z)$ map using the vertical displacement spectra of the tracked reflectors. The results obtained using both methods and models are then compared to check if they are consistent, and to gain confidence in the HR-MCS methodology. We then analyze and discuss the high-resolution 2D maps in terms of mixing.

## 3. RESULTS

The procedure described above allowed producing a smoothly varying $k_\rho(x,z)$ map that covers the whole profile (Fig. 3). The goal is being able to identify features and processes occurring in the transition between the internal wave and the turbulence subranges, such as the intensity and scales of variability of the mixing patches, the location and size of the mixing patches and their potential relationship with oceanographic features such as IWs or wave instabilities. For this, we also use the vertical $k_\rho(z)$ profile obtained from the XCTD and ADCP (Fig. 4a).

### 3.1. Probe-based $k_\rho(z)$ profile

As we mentioned above, to have a reference value to compare with the MCS-based $k_\rho(x,z)$ maps, we have first calculated a $k_\rho(z)$ profile for shallow waters (< *200 m*) using the XCTD and ADCP data and applying the Gregg89 model (Eqs. 4-6). To do this we have used ADCP measures averaged within 10 m-depth bins. By doing this, we obtain an average value for the shear variance of $S_{10}^4 = 0.28\ s^{-4}$, whereas the reference value of the shear variance obtained from the Garrett-Munk model (Gregg, 1989) is $S_{GM}^4 = 0.013\ s^{-4}$. This gives an average dissipation rate $<\varepsilon> \approx 1.3x10^{-8}\ Wkg^{-1}$, and an average diapycnal diffusivity $<k_\rho> \approx 10^{-3.0}\ m^2s^{-1}$ for the targeted depth range (Fig. 4a). The $k_\rho(z)$ profile obtained from the XCTD and ADCP is also shown in Fig. 4a, together with the global averages for overturning ($<k_\rho> \approx 10^{-4}\ m^2s^{-1}$) as well as the average pelagic diffusivity in the ocean ($<k_\rho> \approx 10^{-5}\ m^2s^{-1}$).

We obtain minimum values of the mixing rate at *50-55 m*, *68-73 m*, and *100-125 m*. The absolute minimum of $k_\rho = 10^{-5.2}\ m^2s^{-1}$ is obtained at *~115 m*, whereas the maximum is of $10^{-2.1}\ m^2s^{-1}$ at *~15 m*. This gives a range of variation of $10^{-3.1}\ m^2s^{-1}$. Deeper than this, mixing variability is smaller. The Turner angle and buoyancy frequency (Fig. 4b) indicate that the region is mostly stable with a slight tendency to double-diffusion (*Tu≈45°*).

It is worth noting that, at this specific location, the average vertical $\varepsilon(z)$ and $k_\rho(z)$ values are one order of magnitude higher that the global average ones. The higher values probably reflect the effect of overturning in the thermocline. While probe-based measurements are well suited to investigate mixing variability in the vertical dimension, they do not provide information on the variability in the horizontal dimension with a comparable level of detail. As explained above, to do this we have used estimations of $\varepsilon$ and $k_\rho$ based on the HR-MCS data, but applying Batchelor59 model (Eq. 7) instead.

### 3.2. High-resolution multichannel seismic-based $k_\rho(x, z)$ map

The $k_\rho(x,z)$ map displayed in Fig. 3 has average values of $<\varepsilon> \approx 6.5x10^{-9}\ Wkg^{-1}$ and $<k_\rho>\approx10^{-2.7}$ $m^2s^{-1}$. These values are within the range of values obtained from the XCTD and ADCP data but, at the same time, they are over an order of magnitude higher than the global ocean reference value of $k_\rho \approx10^{-4.0}\ m^2s^{-1}$ (Fig. 4a). Figure 5 displays the $k_\rho(x,\ z)$ map superimposed with the HR-MCS data. It is worth noting that the range of horizontal variability is similar to that observed in the vertical dimension, although there is no direct visual correspondence between the $k_\rho$ anomalies and IWs. The range of variability is of over three orders of magnitude, locally reaching an extreme value of $k_\rho \approx10^{-1.5}\ m^2s^{-1}$ at a depth of ~55 m and at 16 km along the line; and a minimum value of $k_\rho \approx10^{-4.5}\ m^2s^{-1}$ at ~95 m depth and 20 km along the line, which is close to the global oceanic average.

To better illustrate the procedure followed to generate the maps, several examples of $k_\rho$ values obtained in "high" and "low" mixing areas, and the corresponding window average of the computed displacement spectra, are shown in Fig. 6. Numerous patches with $k_\rho$ values exceeding $10^{-2}\ m^2s^{-1}$, with a characteristic size of 1-2 km in the horizontal dimension and ~10 m in the vertical are found throughout the whole section (i.e., the yellowish patches in Figs. 3 and 5). Not only the average depth value, but also the vertical size of the anomalies, as well as the range of $k_\rho$ variation, are in agreement with the probe-based values (Fig. 4). The contribution of the high $k_\rho$ patches to the local average value is therefore outstanding, raising it from a background average value of ~$10^{-4}\ m^2s^{-1}$ to ~$10^{-2.7}\ m^2s^{-1}$.

### 3.3 Correspondence between mixing hotspots and imaged oceanographic features

To discuss the possible origin, or nature, of the mixing hotspots identified in the $k_\rho(x,z)$ map (Fig. 3), we have visually compared the lateral variation of diapycnal diffusivity, with the structures imaged at the different subranges, along several individual reflectors. The analyzed reflectors have been selected as examples of high diffusivity (H1 and H2 in Fig. 7) and low diffusivity (H3 and H4 in Fig. 8) areas. To calculate $k_\rho(x)$ along each horizon we have computed the spectral energy within a 1.2 km-wide window moving laterally 30 m at each step along the whole reflector. To analyze the features that contribute to the energy spectrum in the different scales, and to compare them in turn with $k_\rho(x)$, the horizons have been filtered at the scale ranges attributed to the IW (3000-100 m), transitional (100-30 m), and turbulent (30-13 m) subranges, respectively. As a reference, the local horizontal buoyancy wavelength estimated from the XCTD data is $l_N\approx90\ m$ (Sallares et al., 2016). Although no general conclusions should be extracted from the analysis of a few individual reflectors, they show some relevant trends and correspondences to be taken into account when interpreting the results. In this sense, a clear trend that is observed in the displacement spectra is the systematic steep slope obtained at the transitional subrange between IWs and turbulence (Fig. S3). As explained in Sallares et al (2016), this slope is consistent with numerical estimates for the evolutionary stage of the vortex sheet linked to shear instabilities (Waite, 2011), and it likely reflects the loss of energy in the wave field due to dissipation (e.g. Samodurov et al., 1995).

Regarding horizons crossing high dissipation areas (H1 and H2 in Fig. 7), a striking feature is the correspondence between the amplitude of the vertical displacements imaged in the transitional subrange and the variation in $k_\rho$. Hence, a variation in the amplitude of the features observed in the transitional subrange, at *~34.7* km along horizon H1, and at *~12.4* km along H2 (red lines in Fig. 7a), coincide with a decrease in $k_\rho$. In the case of H1, the average $k_\rho$ value to the left of this point is $10^{-2.5}$ $m^2s^{-1}$, while right of this point, it is $10^{-3.0}$ $m^2s^{-1}$, whereas in the case of H2, the average $k_\rho$ value to the left of this point is $10^{-4.1}$ $m^2s^{-1}$, while right of this point, it is $10^{-2.9}$ $m^2s^{-1}$. Although most of these values are higher than the average global value for meridional overturning circulation, the highest local average values are obtained in the region where the clearest, largest amplitude features, possibly representing KH billows (Sallares et al., 2016), are imaged. While we can identify a visual correspondence of high mixing values and the largest-amplitude features imaged in the transitional subrange, no direct correspondence is found with specific IWs, which are ubiquitous all along the profile.

For the low dissipation areas (Fig. 8), we have selected H3, which is located at *~35 m* depth and has a length of *~3.5 km* (*17.5-21 km* along profile), and H4, located at *~95 m* depth and *~4.0 km-*long (*32.5-36.5 km* along profile). They were selected because their location coincides with a relatively weak mixing area, according to the $k_\rho(x,z)$ map (Fig. 8a). As in the previous case, we have first calculated $k_\rho$ using the spectral energy values within *1.2 km*-wide window, moving laterally *30 m* at each step, along the whole reflector length. In the case of H3, the average value for the whole horizon is $k_\rho \approx 10^{-4.2}$ $m^2s^{-1}$, so considerably lower than in H1 but close to the global average value, whereas for H4, it is $k_\rho \approx 10^{-4.1}$ $m^2s^{-1}$. In this case, we have identified some peaks at the transitional subrange that coincide with local highs in $k_\rho(x)$. The peaks in H3 (at *18.4 km, 19.3 km, 19.9 km and 20.4 km*) and in H4 (at *33.5 km, 34.9 km, 35.3 km, and 35.8 km*) display higher $k_\rho$ values than the global ocean average. In particular, we obtain $k_\rho \approx 10^{-3.1}$ $m^2s^{-1}$, $k_\rho \approx 10^{-3.3}$ $m^2s^{-1}$, $k_\rho \approx 10^{-3.4}$ $m^2s^{-1}$, and $k_\rho \approx 10^{-3.5}$ $m^2s^{-1}$, respectively, for each peak in H3, and $k_\rho \approx 10^{-3.0}$ $m^2s^{-1}$, $k_\rho \approx 10^{-3.2}$ $m^2s^{-1}$, $k_\rho \approx 10^{-3.1}$ $m^2s^{-1}$, and $k_\rho \approx 10^{-2.9}$ $m^2s^{-1}$, respectively, for each peak in H4. There are four more peaks at *19.5 km* and *20.1 km* in H3 and *34 km* and *36.3 km* in H4 that show no visual correspondence with structures in the transitional subrange but with larger amplitude features in the turbulent subrange, so we hypothesize that they could be related to smaller-scale turbulent processes. The segments with no direct visual correspondence with $k_\rho$ peaks, instead, display average $k_\rho \approx 10^{-4.3}$ $m^2s^{-1}$, which is close to the global ocean average (Fig. 4a).

## 4. DISCUSSION

The spatial variability observed along isopycnals based on the spectral analysis of the seismic data allows identifying a number of local features at different evolutionary stages. These features are the manifestation of relevant oceanographic processes and structures, such as internal waves in the IW subrange, hydrodynamic instabilities in the transitional subrange, and turbulence at smaller scales. These processes are likely responsible for disruption of finestructure in the seismic image, and the high amplitude variability or the abrupt fading of some reflectors.

The large variations observed in the $k_\rho(z)$ profile (Fig. 4), together with the slight tendency to double-diffusion identified in the Turner angle, suggest that the system is prone to be affected by advection processes (e.g. Kunze and Sanford, 1996). Mixing appears to concentrate within the MAW, where the shear is strongest in the study area, and not deeper than > *110 m*, where there is

no significant shear and the system is weakly stratified. The shear to strain ratio calculated applying the Gregg89 model ($S_{10}^4/S_{GM}^4 = 21$), indicates that the energy in the IW field is higher than that of the GM model, for which $S_{10}^4/S_{GM}^4$ usually $\approx 3$. We can therefore make the assumption that the energy is distributed in the whole inertial range where the water structures are stable (e.g. Munk, 1981). Similar results were obtained by Holbrook et al. (2013), who registered a shear to strain ratio of *17*. The IWs can therefore be considered as an energy distributor from anisotropic to isotropic motions. The $k_\rho$ value obtained from XCTD and ADCP using Gregg89 model is $k_\rho \approx 10^{-3.0}\,m^2s^{-1}$, whereas we obtain $k_\rho \approx 10^{-2.7}m^2s^{-1}$ using MCS data and the Batchelor59 model. The ranges of variation in the two cases are also comparable, from maximum values of *$10^{-2.2}m^2s^{-1}$* and *$10^{-1.5}m^2s^{-1}$*, to minimum values of *$10^{-5.4}m^2s^{-1}$* and *$10^{-5.7}m^2s^{-1}$*, respectively, for the two methods. These similar values obtained based on different models and using independent techniques are well above the global average, suggesting that the downward energy cascade to small scales is highly efficient in the surveyed area.

We find no clear correlation between local $k_\rho$ variations and the presence of individual IWs, which are clearly imaged and display a rather homogeneous distribution all along the line. Conversely, we find some hints of a direct relationship between changes in the amplitude of vertical isopycnal displacements and variations in $k_\rho$ (Figs. 7 and 8), or between local peaks in the amplitude of vertical displacement and high $k_\rho$, in the transitional domain. Our interpretation is that IW-induced mixing is probably not efficient enough to sustain the overturning in the target area (Figs. 7 and 8). Instead, the correspondence between high-amplitude features in the shear instability-dominated transitional domain with $k_\rho$, suggests that the energy transfer between IWs and turbulence is probably enhanced by shear instabilities. The lack of clear correlation between IWs and turbulence agrees with Klymak and Moum's (2007a) assumption, suggesting a weak dependence of mixing rates on IW energy. Our observations indicate that the processes of IW destabilization and breaking appear to be important to allow transferring energy towards smaller scales efficiently and enhance mixing. In the case of the Alboran thermocline, the main mechanism could well be the development of shear instabilities, but other processes such as the interaction of IWs with rough bathymetry could also play an equivalent role in other regions and settings (Dickinson et al., 2017).

The "mixing hotspots" identified in the $k_\rho(x,z)$ map (Fig. 3), likely represent a significant source of regional diapycnal mixing at the boundary layer between the MAW and the MW (*30 – 200 m*), which is subject to vertical stratification and shear values of *$3.2x10^{-3}\ s^{-1}$*. The mixing and energy transfer between these two water masses constitutes the main energy source of the region. In contrast to other regions (e.g. the Gulf of Mexico as in Dickinson et al., 2017), the smooth and relatively deep seafloor along the profile (> *800 m* in average; Fig. S4c) suggests a small contribution of interaction with bathymetry in the generation of the mixing hotspots. Given that the MAW-MW boundary layer is subject to shear (Fig. S4a, and S4b), and taking into account the visual correspondence between the location of the largest amplitude features in the transitional domain and high mixing values along individual reflectors, we hypothesize the existence of a direct link between mixing hotspots and IW shear instabilities. This could explain the high mixing variability throughout the surveyed area despite the ubiquitous presence of IWs.

The $k_\rho$ values along H1 exceed the global average for overturning along most of the reflector ($<k_\rho>$ $\approx 10^{-2.5}\ m^2s^{-1}$), with lower values only at some specific locations (Fig. 7b). The spatial correspondence between high diffusivity values and the presence of large-amplitude features at

the transitional subrange, interpreted to correspond to KH-like shear instabilities (Sallares et al., 2016), is conceptually equivalent to the mechanism proposed by Gregg (1987), where mixing at the transitional subrange occurs principally at vortex sheets through wave-instability. As an example, we hypothesize that in horizon H1, the presence of a vortex sheet left of ~*35* km produces the high mixing values, whereas to the right there is no vortex sheet and the ocean is more stable. The correspondence between diapycnal peaks and wave-instabilities in horizon H3 suggests a similar situation (Fig. 8). Our results indicating a patchy ocean interior coincide with those previously presented by Sheen at al. (2009), Fortin et al. (2016), and Dickinson et al. (2017), but our work allows extending the conclusions to smaller scale-processes and shallower ocean levels. Additionally, we identify the development of shear instabilities as a likely relevant mechanism driving the downward energy cascade between IWs and turbulence at the thermocline depth that should be taken into consideration in ocean dynamic models.

## 5. CONCLUSIONS

We have used acoustic images obtained with a high-resolution MCS system to produce a 2D diapycnal mixing map around the thermocline of the Alboran Sea. Our results confirm a high level of diapycnal variability and the presence of marked mixing hotspots in the water column. The $k_\rho$ *(x,z)* map obtained by applying the Batchelor59 model to the seismic data, has a strong variability with values ranging between $<k_\rho> \approx$ *$10^{-1.5}$ $m^2 s^{-1}$*, in the brightest hotspots, and $< k_\rho > \approx$ *$10^{-3.3}$ $m^2 s^{-1}$*, in the background. The obtained values are high enough to account for overturning at thermocline depths. The mixing hotspots have a characteristic size of *10-15 m* in the vertical dimension, and *1-2 km* in the horizontal one, although there are also some smaller-size ones. They are located at different depths within the thermohaline layer, although they appear to concentrate in highly sheared regions. The comparable values obtained with independent XCTD- and ADCP-based measures, confirm that HR-MCS is a useful technique to study processes and structures occurring at the sub-mesoscale, which are difficult to capture and characterize by other means.

We investigate the relationship between mixing variability and ocean dynamics at different spatial scales by analyzing the spectral amplitudes along four seismic horizons in the internal waves and transitional, or instability-dominated, subranges. On one hand, we found no clear correspondence between the location of the mixing patches and the location and amplitude of individual IWs, which are imaged all along the surveyed area. Conversely, a visual correspondence exists between the location of high-amplitude isopycnal vertical displacements at the instability-dominated transitional subrange and high mixing values in different reflectors, suggesting a causal relationship between both features. We interpret the development of shear instabilities as a mechanism that locally enhance downscale energy transfer between IWs and turbulence. Areas displaying the most vigorous instabilities coincide with the highest estimated diapycnal mixing values, which are well above the global average value for meridional overturning. This observation suggests that the energy transfer from anisotropic to isotropic scales is highly efficient at thermocline depths within the studied area.

Overall, our study shows that the HR-MCS technique can be used to study sub-mesoscale structures and processes at the thermocline level, provided that the stratification is strong enough to produce acoustic reflectivity that can be recorded by the system. The high-resolution 2D maps produced from the seismic reflectivity could help improving estimates of the parameters to be incorporated in numerical models of ocean dynamics.

**ACKNOWLEDGEMENTS**

This work has been fulfilled in the framework of projects POSEIDON (Ref: CTM2010-25169) and APOGEO (Ref: CTM2011-16001-E/MAR), both funded by the Spanish Ministry of Economy and competitiveness (MINECO). The seismic and oceanographic data were acquired in the framework of the IMPULS survey (Ref: 2003-05996-MAR) also from MINECO, and SAGAS survey (Ref: CTM2005-08071-C03-02/MAR-SAGAS). Helpful comments were provided by Josep Lluis Pelegrí, Miguel Bruno, numerous colleagues at the Barcelona-CSI, and Diana Francis and David M. Holland from the Center for Global Sea Level Change (CSLC) – NYUAD, Abu Dhabi, UAE.

**APPENDIX A**

Table A1. Parameters used in text

| Variable | Value | Description |
|---|---|---|
| $f$ | $0.00008613$ s$^{-1}$ | Coriolis f. at 36° |
| $N$ | 5 cph = $0.00138$ s$^{-1}$ | Buoyancy frequency (ocean average) |
| $V$ | $0.207$ m s$^{-1}$ | RMS amplitude of velocity fluctuations |
| $\nu$ | $0.000001064$ m$^2$ s$^{-1}$ | Kinematic Viscosity |
| $C_T$ | $0.4$ | Proportionality constant |
| $\Gamma$ | $0.2$ | Empirical value of mixing efficiency (Osborn and Cox, 1972). |

**APPENDIX B**

Buoyancy Reynolds number

Gargett et al, (1988) use an index to know if the system is isotropic or not, allowing to determine if the buoyancy flux is vigorous enough to generate turbulence and therefore a high mixing level (Thorpe, 2005). The index depends on kinematic viscosity and is called Buoyancy Reynolds number:

$$R_B = \varepsilon / \nu N^2 \qquad \qquad \text{(B1)}$$

The mean kinematic viscosity in the ocean is $\nu = 1 \times 10^{-6}$ m$^2$ s$^{-1}$. Some properties of the inertial subrange are consistent with isotropy for values of $R_B < O(10^2)$. To consider anisotropy and avoid serious underestimates of mixing, Smyth and Moum (2000) propose $R_B > 200$ as safe for high mixing levels due to free viscous effects. For our submesoscale regime $R_B = 3200$, a value that is compatible with the calculated mixing levels.

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

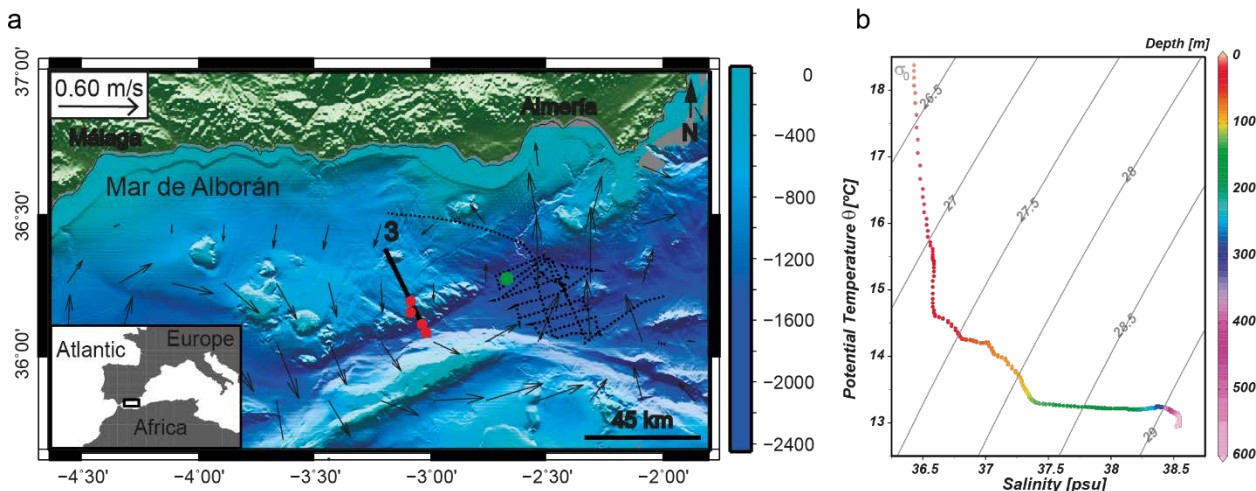


Figure 1. (a) Bathymetric map of the Alboran Sea and location of the data used in the study. HR-MCS profile acquired during the IMPULS-2006 experiment (black line labelled 3), eXpendable Bathy-Thermograph (XBTs) profilers (red circles), eXpendable Conductivity Temperature Depth (XCTD) probe (green circle). Acoustic Doppler Current Profiler (ADCP) lines (black dotted line).

Geostrophic velocity for May 17th, 2006 (gray arrows). (b) Temperature-Salinity diagram from XCTD probe. $\sigma_0$ is the potential density in kg/m$^3$. Color scale indicates depth.

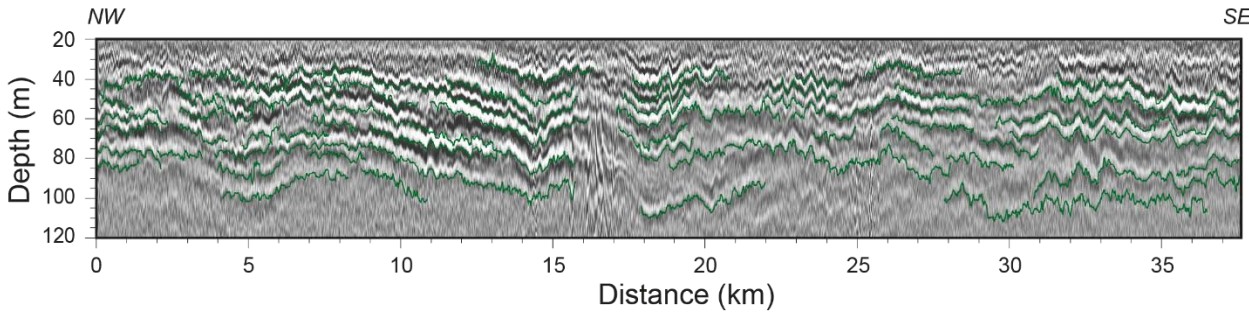

Figure 2. Depth-converted high-resolution multichannel seismic profile, with the tracked reflectors
superimposed (green lines). (See Fig. 1a for location).

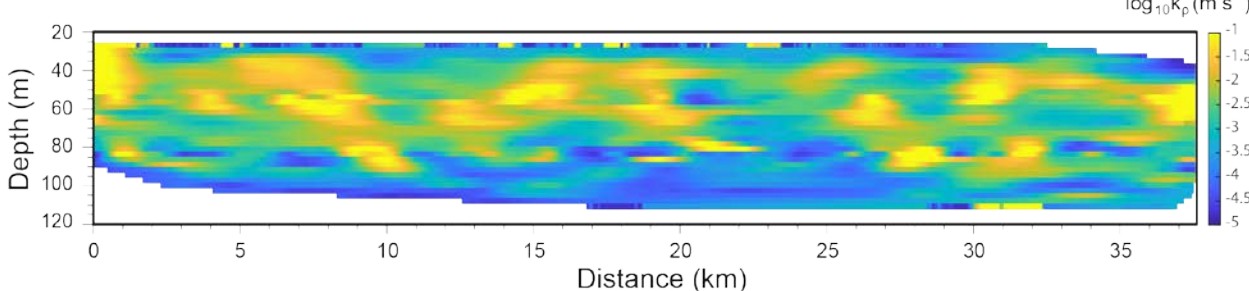

Figure 3. $k_\rho(x, z)$ map obtained along the seismic profile indicated in Fig.1, following the procedure explained in the text. White colored areas correspond to poorly sampled areas, with too few data to properly calculate $k_\rho$.


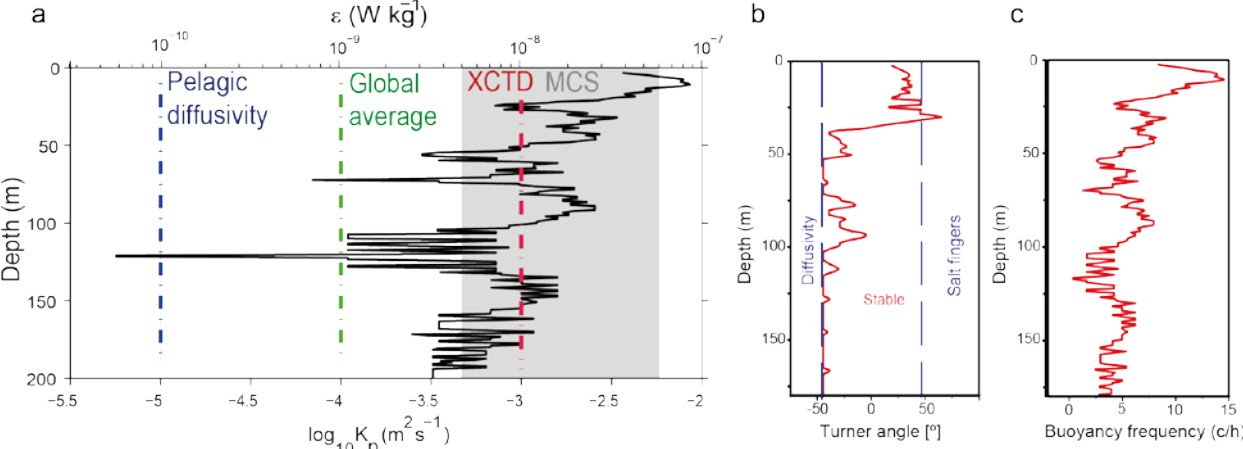

Figure 4. (a) Depth profile of $\varepsilon(z)$ and $k_\rho(z)$ obtained from XCTD and ADCP data and applying Gregg89 model. The blue dotted line is the pelagic diffusivity in the ocean ($k_\rho \approx 10^{-5} \, m^2 s^{-1}$), the green dotted line is the global average for overturning ($k_\rho \approx 10^{-4} \, m^2 s^{-1}$), the red dotted line is the average vertical profile from XCTD and ADCP data ($k_\rho \approx 10^{-3.0} \, m^2 s^{-1}$) and the gray area is the incidence range from MCS data ($k_\rho \approx 10^{-2.7} \, m^2 s^{-1}$). (b) Turner angle showing ranges, the blue dotted lines shows where the water column is unstable to diffusivity ($Tu < -45°$), stability ($-45° < Tu < 45°$) and prone to salt fingering ($Tu > 45°$), and (c) buoyancy profile calculated with the XCTD data.

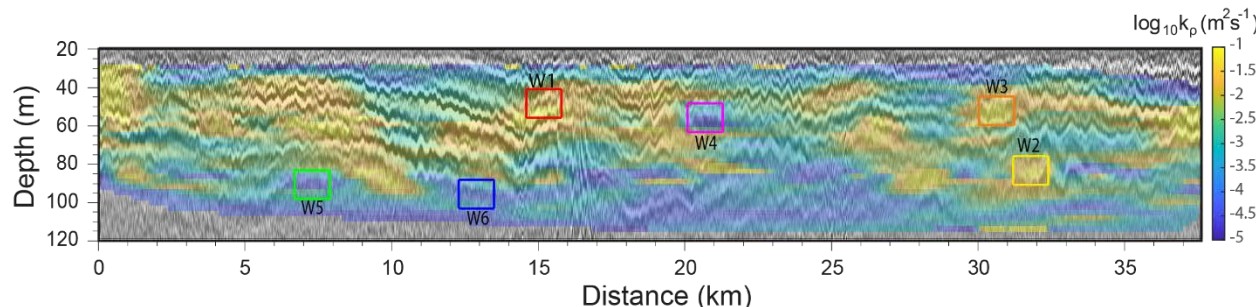

Figure 5. High-resolution $k_\rho(x,z)$ map overlapped with the HR-MCS image. Squares indicate location of some of the *1200 m* x *15 m* windows analyzed. They have been selected as examples of high-dissipation (windows W1-W3) and low-dissipation (windows W4-W6) areas. The color code of the squares is the same as for reflector spectra in Fig. 6, so that colors coincide with those of displacement spectra within the corresponding window.


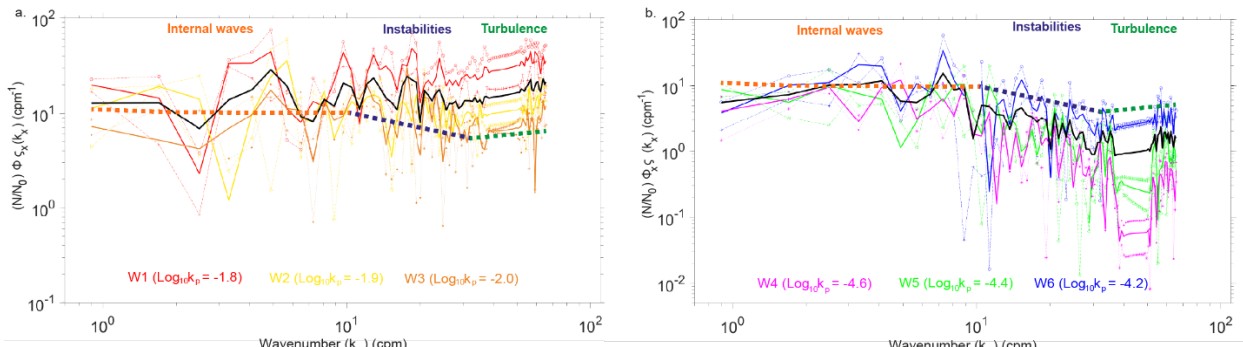

Figure 6. Average horizontal spectrum of the vertical displacements of reflectors inside windows
W1-W6 (see location and color code in Fig. 5). (a) Spectra of individual reflectors in "high
diffusivity" areas (thin dotted lines), average within windows W1 (red solid line), W2 (yellow
solid line), and W3 (orange solid line), and average of the three "high diffusivity" windows (thick
solid black line). (b) Spectrum of individual reflectors in "low diffusivity" areas (thin dotted lines),
average within windows W4 (magenta solid line), W5 (green solid line), and W6 (blue solid line),
and average of the three "low diffusivity" windows (thick solid black line). The reference lines are
the theoretical slopes corresponding to the GM79 model for the internal wave subrange (brown
dotted line), Kelvin-Helmholtz instabilities for the transitional subrange (dark blue dotted line),
and Batchelor59 model for turbulence (dark green dotted line). Legend: Values of diapycnal
diffusivity using spectral values at the turbulent subrange within each of the analyzed windows
(same color code as for windows W1-W6).

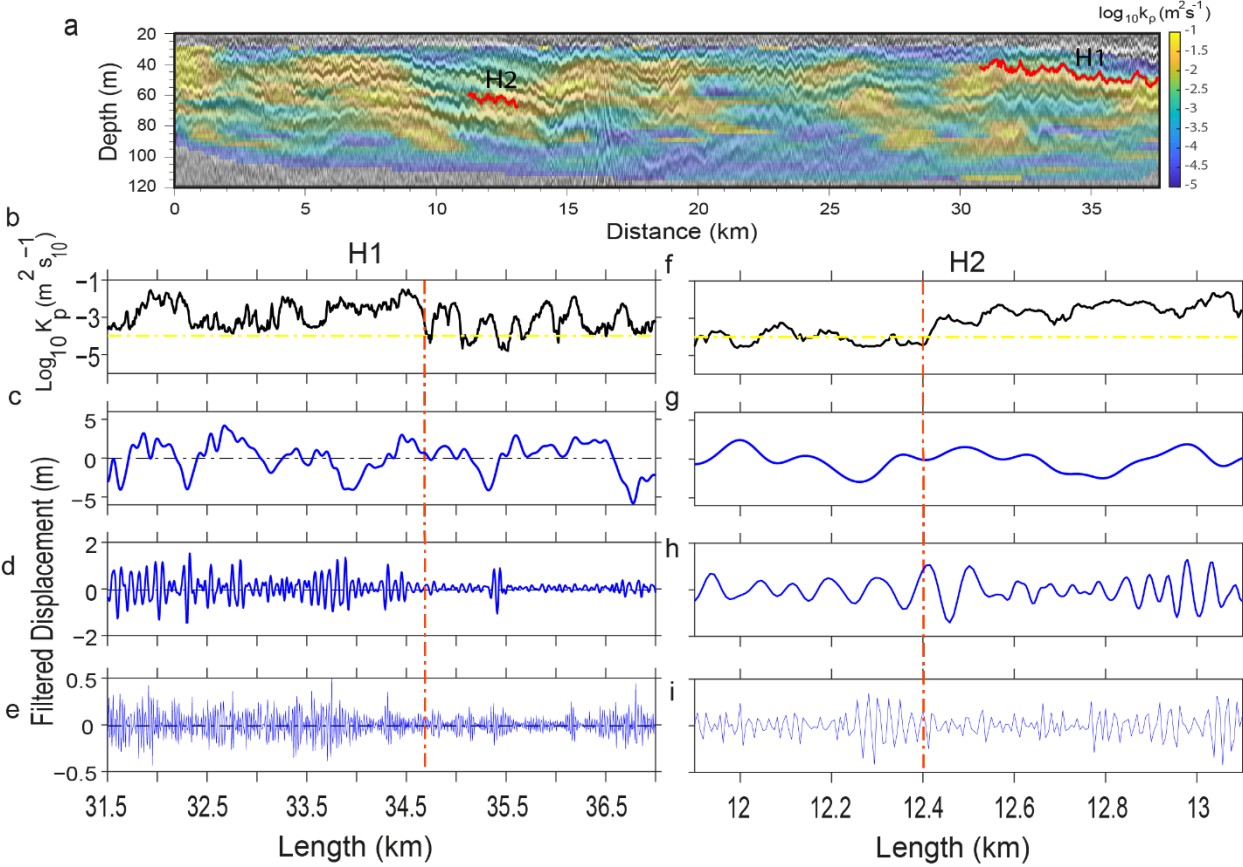

Figure 7. (a) High-resolution $k_\rho(x, z)$ map overlapped with the HR-MCS image. Red lines indicate location of 2 horizons analyzed (H1 and H2). They have been selected as examples of reflectors crossing higher mixing areas. (b). Diapycnal mixing obtained along H1 (see details of calculation in the text). (c) Signal filtered at wavelength ranges of the IW sub-range (*3000-100 m*), (d) the transitional subrange (*100-30 m*), and (e) the turbulent subrange (*30-13 m*). The dashed red lines identifies the "breaking point" referred to in the text. (f, g, h, i) same as (b, c, d, e) for H2.

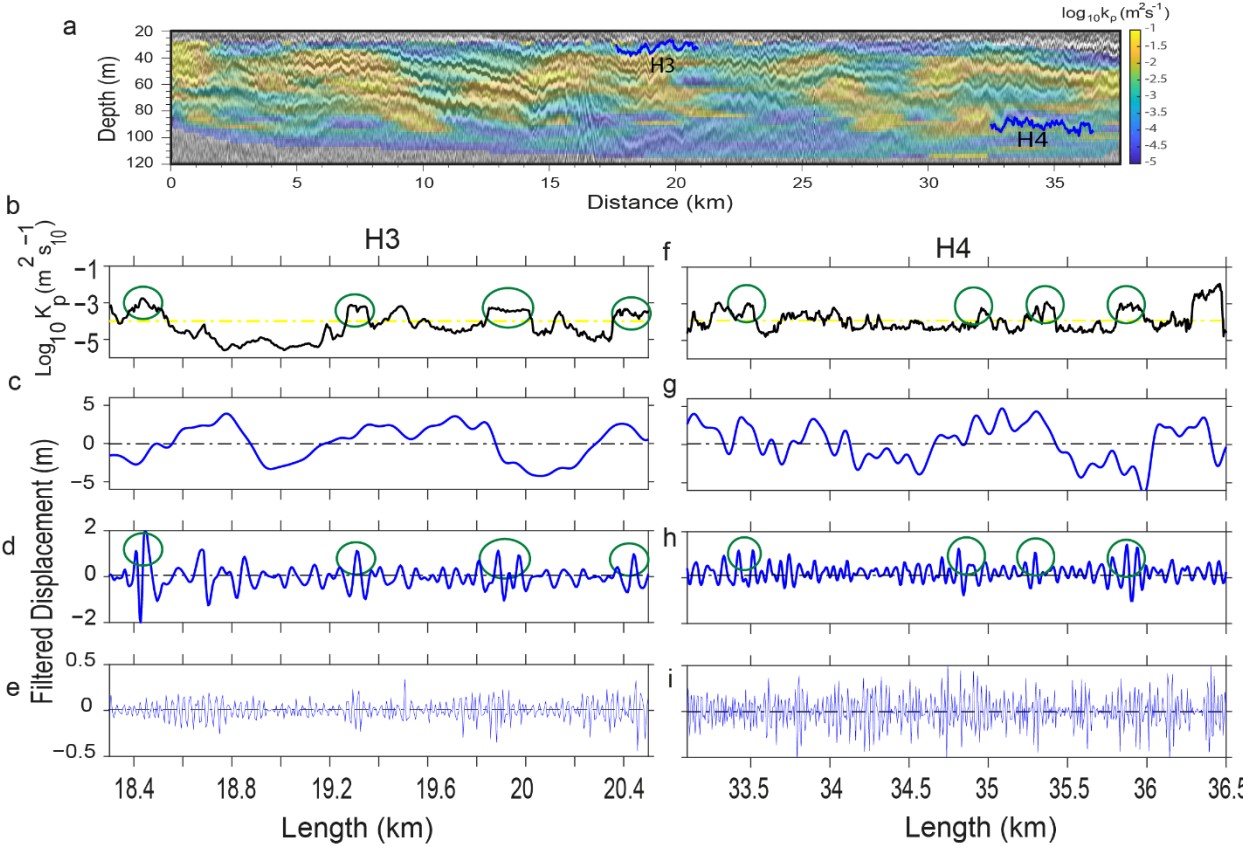

Figure 8. (a) High-resolution $k_\rho(x, z)$ map overlapped with the HR-MCS image. Blue lines indicate location of 2 horizons analyzed (H3 and H4). They have been selected as examples of reflectors crossing lower mixing areas. (b). Diapycnal mixing obtained along H3 (see details of calculation in the text). (c) Signal filtered at wavelength ranges of the IW sub-range (*3000-100 m*), (d) the transitional subrange (*100-30 m*), and (e) the turbulent subrange (*30-13 m*). The green circles identifies the "diffusion peaks" referred to in the text. (f, g, h, i) same as (b, c, d, e) for H4.

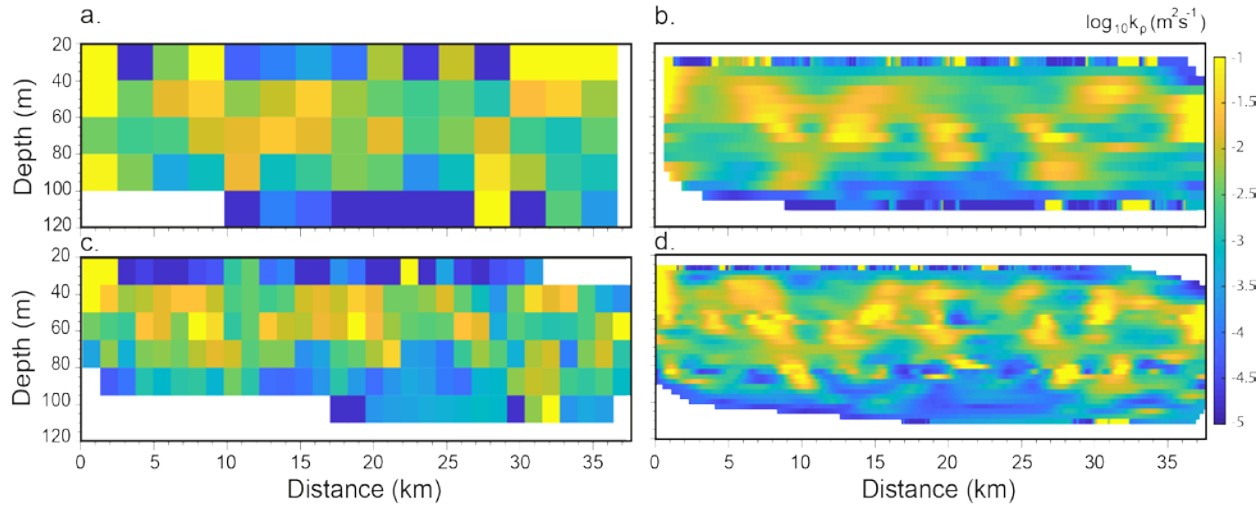

Figure S1. Four examples of $k_\rho(x, z)$ maps obtained along the seismic profile shown in Fig.1, using windows of different size, with and without applying the sliding window approach. (a) Window size is *2400 m* wide x *20 m* high. (b) Same window size as in (a), but applying sliding window step of *60 m* in the horizontal and *6 m* in the vertical one, between neighboring windows. (c) Window size is *1200 m* wide x *15 m* high. (d) Same window size as in (c), but applying a sliding window step of *30 m* (horizontal) and *3 m* (vertical). This is the one selected and used for the analysis.

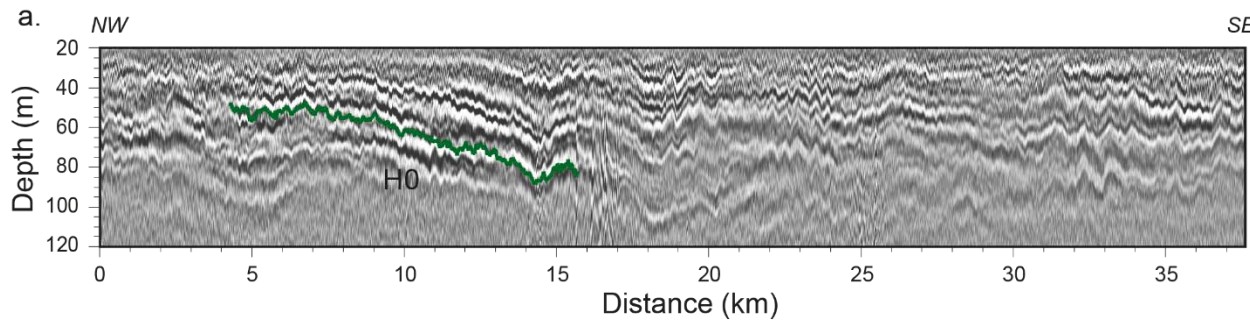

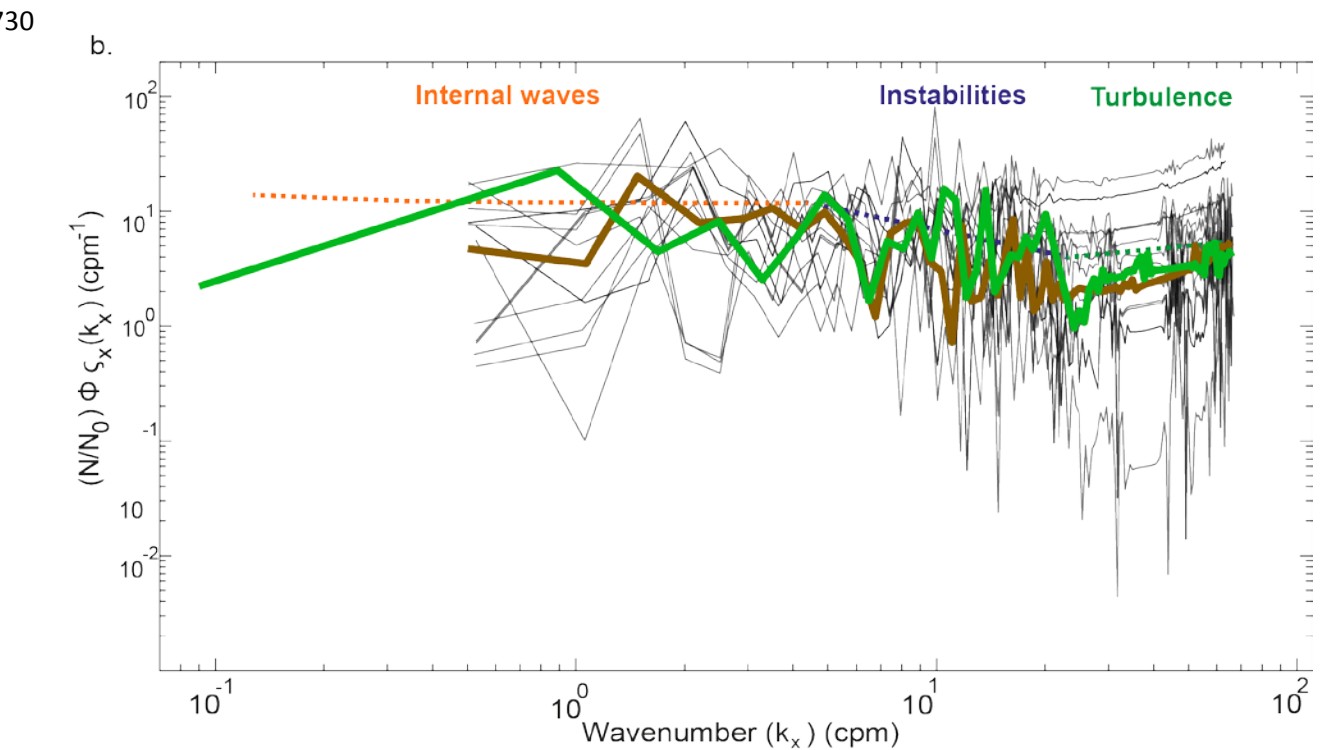

Figure S2 (a) Depth-converted high-resolution multichannel seismic profile (Here we show a new horizon H0, green line). (b) Horizontal spectrum of the vertical displacement of reflector H0. (green line) considering the whole reflector. (black lines) spectrum from the reflector split in ten 1.2 km-long segments. (brown line) average spectrum from the 10 segments. Segments, the average and the whole reflector show the same trends in the scales of interest.

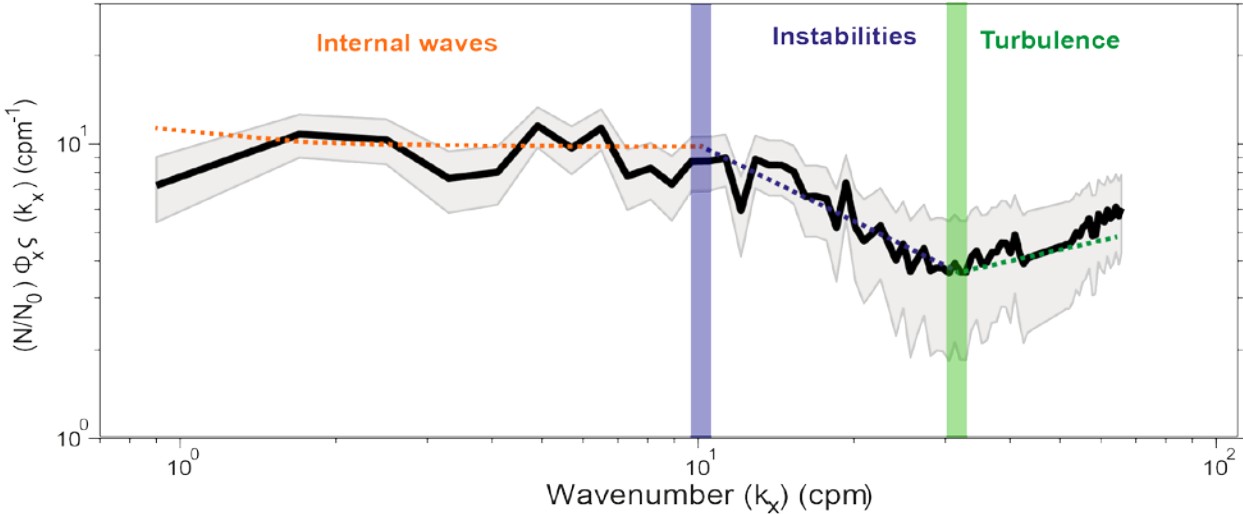

Figure S3. Average horizontal spectrum of the 68 tracked reflectors scaled by the local buoyancy frequency at the reflector depth, and multiplied by $(2\pi k_x)^2$ (solid line) and its corresponding 95% confidence interval $(2\sigma)$ (shaded area). The reference lines are the theoretical slopes corresponding to the GM79 model for the internal wave subrange (red line), Kelvin-Helmholtz instabilities for the transitional/buoyancy subrange (blue line), and Batchelor59 model for turbulence (green line). The steeper slope at the highest wavenumbers corresponds to noise. The blue rectangle marks the buoyancy scale ($l_N \approx 100 \, m$), and the green rectangle the limit between transitional and turbulent subranges ($\sim 30 \, m$).

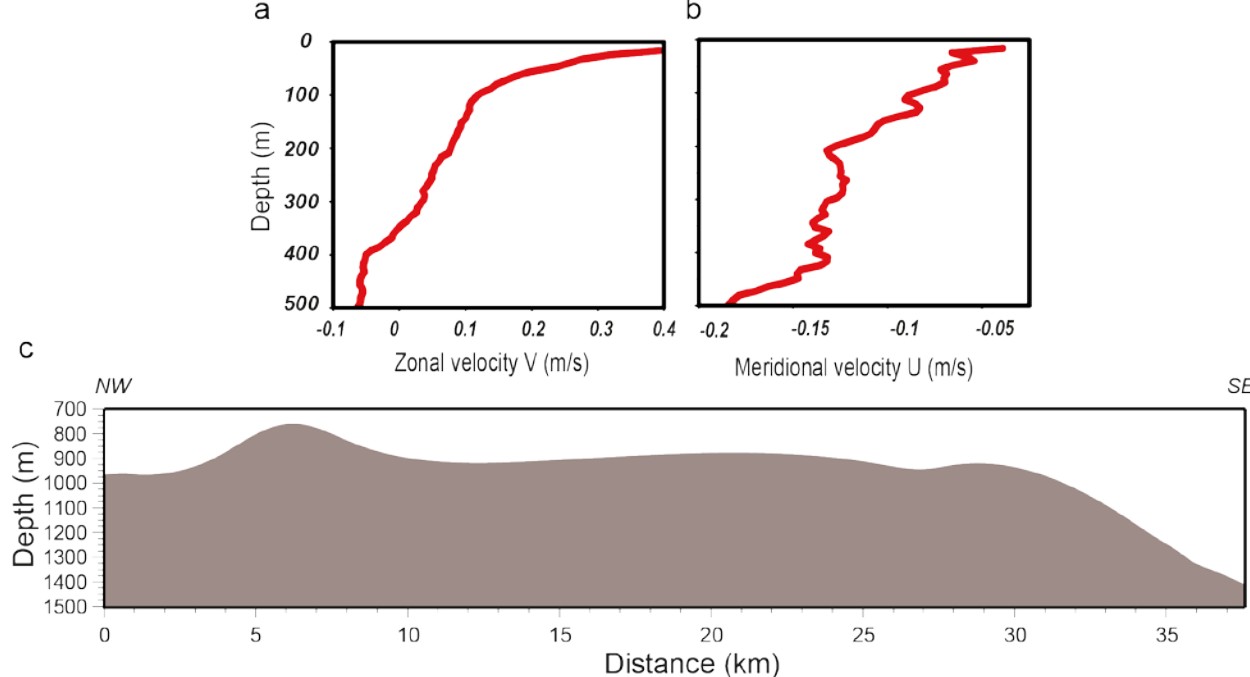

Figure S4. (a) Current velocity profile from ADCP data, SAGAS in May, 2010. (a) The zonal velocity ($V$) variations, and (b) the meridional velocity ($U$) variations according to the depth. (c) Bathymetric profile over the seismic profile.