# Peer review of "High-resolution diapycnal mixing map of the Alboran Sea thermocline from seismic reflection images"

_Ocean Science, 2017_

## Referee Comment (RC1) · Anonymous Referee #1 · 10 Oct 2017

General Comments:

The paper presents some nice data acoustic data from the Alboran Sea, for which turbulent mixing estimates within the thermocline have been estimated at high spatial resolutions. mixing estimates show an interesting 'patchyness'. Some oceanographic data has also been used to compare mixing rate estimates. Overall the paper is well put together and the data well presented. However, I feel that the paper requires some further work on three counts: 1 - Clarification should be provided regards the physical mechanisms behind the mixing distribution observed. Messages regards the influence of internal waves seem confused 2 - The analysis of mixing-rates from oceanographic

data, using fine-structure estimates (and assumptions therein) needs to be improved - see comments below 3 - I would like to see a comparison of mixing estimates form the internal wave and Batchelor regimes of the MCS spectra

Specific comments: Lines 35-37: Please clarify here - I think that both your references refer to internal-wave phenomena

Lines 50-53: I am not sure what you are saying here.

Lines 65: I think that lowered microstructure profiles are generally the most robust source of turbulent measurements.

Line 150: Do you have evidence for this? Could you compare acoustic reflection horizons to density horizons in the oceanographic data?

Line 165: If possible you should use integrated shear and or strain spectra to get estimates from CTD/ADCP data - perhaps you are limited by depth ranges? You are also missing some terms in for shear-to-strain ratio and inertial frequency e.g. see Waterman, S., K. L. Polzin, and A. C. Naveira-Garabato (2012), Internal waves and turbulence in the Antarctic Circumpolar Current, J. Phys. Oceanogr., 43, 259–282. You should at least quantify the omission of these terms and also explain how you decide on what you mean by 'uncertainty bounds' in several places in the text. You should also mention the errors associated with fine-structure estimates - particularly in regions away from the open ocean.

Lines 195-200: This should be in methods

Line 203: Spatial resolution - be careful what you mean by this as really each data point is an average over 1200m by 15 m box

Lines 215: What scale are you computing shear over i.e. dZ? Also how to you quantify buoyancy frequencies, N?

Lines 319: Shear to strain ratios tell you about the frequency content of the internal

wave field. You might well expect higher inertial content (i.e high shear to strain ratios) near the surface due to wind generation.

Lines 325: How do dissipation estimates compare for GM and Batchelor parts of the MCS spectra? This may tell you something about the role of IW in generating the turbulence/GM assumptions

Line 334 and 351: Confusing regards what you are trying to report regards role of internal waves here - please clarify.
* * *

---

## Referee Comment (RC2) · Anonymous Referee #2 · 11 Oct 2017

This manuscript presents seismic reflection data from the Alboran Sea and outlines a method for producing a map of diapycnal diffusivity across one profile. The major conclusion is that the profile shows patchy turbulence on the scale of a few kilometers horizontally and 10-15 meters vertically. Further, the authors observe greater mixing in areas of internal wave instability. Results are compared to estimates of turbulence made from XCTD and ADCP data as well as background reference models. Additional analyses of filtered slope spectra examine the relationship between diapycnal mixing and the assigned internal wave and transitional subranges.

The introduction and background are clearly written and well presented. However,

the manuscript takes on the substantial challenge of developing a new method and presenting scientific conclusions at once, and in a relatively short format. As a result, I think many issues addressing the methods, presented data, clarity of conclusions, uncertainties, and reach of results are insufficient.

Major Concerns: Data handling and methods are insufficiently explained. The authors need to be clearer about the stated resolution. It is not accurate to apply a 1200x15 m grid to a 30x3 m grid and claim improved resolution. Many of the 30x3 m cells will not have tracks in them. In fact, at CMP spacing of 7.5 m, you can only have 4 or 5 traces represented (depending how you treat them) and few realistic and meaningful spectra can be taken at that scale. The authors need to explain what happens when tracks are longer than the 1200 m box. They state the tracks are 1.5-21 km long, so no tracks would fit inside the 1200 m grid length and position vertically is also unaddressed. Each track would be included in hundreds of 30x3 m grid cells which seriously undermine any claims at resolving patchy turbulence at their stated resolution.

The authors need to show more of the data that support their methods and conclusions, particularly the reflector tracks and many more spectra. To establish this as both a methods paper and support their science conclusion, these data must be shown and clear. First, show the 68 reflector tracks and discuss their distribution, including why application of a k value obtained on the large scale can be applied to the small scale and how to handle regions lacking tracked reflectors. Second, slope spectra rely on the aggregate data of many tracks to have statistically characteristic behaviors (Klymak and Moum, 2007 part II). All of what we are shown are single-track spectra. Third, more justification of the horizontal wavenumber bounds for the sub-regimes (IW, instability/transition, and turbulent) are needed to illustrate these are accurate bounds for sub-regimes all over the 2D line.

The authors need to be clearer about the role of basic oceanographic features and the expression of turbulent structures. Lines 245-246 state "there is no clear visual correspondence between the k anomalies and the most obvious of the imaged oceano-

graphic features such as IWs" which seems to be against the main thrust of the paper that sub-mesoscale features can be examined through their turbulent expressions, particularly lines 19-21 in the abstract as well as a few points in section 4.

The manuscript thesis is about ocean mixing, as such, the turbulent subrange should be examined for tracks H1 and H2. Perhaps there are corresponding traits between the IW subrange and turbulent subrange, or the transitional subrange and turbulent subrange that may clarify the expression of turbulence by mesoscale and sub-mesoscale oceanographic features.

Uncertainties are problematic and under addressed. In the abstract (line 23) and conclusion (line 372) results are stated to be within uncertainty bounds but nowhere in the data do the authors show or discuss any uncertainty assessments. In section 3.2 uncertainty is briefly mentioned, but again, is simply stated that values are within uncertainty bounds. If the conclusions are supported within an uncertainty range, please show and elaborate.

The major conclusions for the filtered spectra analysis are under supported. Lines 378-381 deliver major conclusions about the relationship between IWs and overturning as well as shear instabilities and mixing hotspots. From the data presented, it appears these conclusions are drawn from the analysis of 2 tracked seismic reflections, H1 and H2. This overstates what is observed in the data, particularly when those two tracks were chosen as end-member individuals chosen for their position in "anomalously high (H1) and low (H2) mixing patches" (lines 264-265).

General Comments: The manuscript thesis states that authors produce a map of diapycnal mixing that show patchy nature. However, they often refer to an average k as a benchmark to compare to conventional methods. The authors need to discuss why they would average the entire map when the main thrust of the paper is that it is heterogeneous. Additionally, the authors need to justify why just 1 XCTD and 1 ADCP data set would accurately reflect the average for their produced mixing map. For example,

even if the XCTD was collected concurrently, what would the implication be if it was dropped at 8 km where k is high, or at 22.5 km where k is low?

Please explain how signal-to-noise is calculated. A signal-to-noise ratio of higher than 8 with 6-fold data is surprising.

Much of section 3 (first paragraph) should be expanded and put into section 2

Please explain why tracks H1 and H2 were analyzed with 1 km windows when the turbulent maps were analyzed with 1.2 km windows. The difference in these windows change the wavenumber range of the IW spectra and might aid some of the confusion around the handling of internal waves in the manuscript. Further, why limit the analysis to ~1 km? If they exist, larger IWs should carry even more energy and may be important.

At many points the authors state there are "no clear correlation" or similar language between filtered spectra (e.g. lines 287, 329-331). Were correlations and statistics taken for each of the 68 tracked reflectors to support, or not, a relationship between the filtered spectra? If so, this should be a major point of the paper and have supporting figures.

In figure 6, please show the actual fit lines for these data. This would allow for a brief discussion in the text of how you calculate spectral energy levels beyond a reference to Sallares et al. 2016.

Specific comments: Line 85: Citation should be Holbrook et al., 2003. Line 175: Authors need to state and explain their choice for b the scale factor Lines 329-330: Suggest rewording. The authors infer IW-induced mixing is not efficient enough to keep the overturning in this dataset, I do not think the data shown makes it clear, particularly on a global scale. Line 340: Suggest rewording. A smooth seafloor likely suggests a lesser role in the generation of hotspot mixing, if it is disregarded entirely, please explain.

---

## Author Comment (AC1) · 1 Nov 2017

Mojica et al. Anonymous Referee 1

We thank reviewer1 for the constructive comments and review that has surely helped to improve the manuscript. We have taken all comments and suggestions into account as indicated in our point-by-point answers below. To clarify our answers, we add a file (Review1-answer1.pdf) as a supplement, easily to read for you.

[Figure]

We agree that the relationship between observed oceanographic features and mixing distribution was unclear. Our main message is that there is not a clear correspondence between the location of IWs (> 100 m horizontal scale) and mixing hotspots, but rather between mixing hotspots and the location of large-amplitude features in the transitional domain (30-100 m horizontal scale). Based on this analysis as well as on previous results presented in Sallares et al (2016), we interpret that these large-amplitude features are the expression of shear instabilities (e.g. KH-type billows). This means that there is not a direct relationship between IWs and mixing. It tends to concentrate where IWs become unstable and instabilities develop, leading to turbulence. We clarify this message in the new version of the text (line 19-21, line 339-341, line 383-386).

We have estimated the $k(x,z)$ map for internal waves and Batchelor regimes (figure rev.1-1). The lower mixing values produced by IWs as compared to the Batchelor regime are clear.

Specific comments: Lines 35-37: Please clarify here - I think that both your references refer to internal-wave phenomena

Thanks for the comment, references have been corrected (line 38).

Lines 50-53: I am not sure what you are saying here. We just want to describe the behavior of $\varepsilon$ in a conservative flow. We have modified the text to make this point clear (line 51-52).

Lines 65: I think that lowered microstructure profiles are generally the most robust source of turbulent measurements. We agree, of course. Our point here was to note that these devices (VMP, microriders), provide measurements in just one dimension (either horizontal or vertical), but seismic data cover both dimensions at once. It is obviously with poorer resolution than microstructure profilers, but much better than that of conventional probe-based studies in the horizontal one (line 63-65)

Line 150: Do you have evidence for this? Could you compare acoustic reflection horizons to density horizons in the oceanographic data? We do not have direct evidence for this particular profile because we cannot invert density with this data and we do not have the appropriate complementary oceanographic data. However, in a previous study by our group with appropriate data, we showed that seismic reflectors do actually follow isopycnals (Biescas et al, 2014).

Line 165: If possible you should use integrated shear and or strain spectra to get estimates from CTD/ADCP data - perhaps you are limited by depth ranges? You are also missing some terms in for shear-to-strain ratio and inertial frequency e.g. see Waterman, S., K. L. Polzin, and A. C. Naveira-Garabato (2012), Internal waves and turbulence in the Antarctic Circumpolar Current, J. Phys. Oceanogr., 43, 259–282. You should at least quantify the omission of these terms and also explain how you decide on what you mean by 'uncertainty bounds' in several places in the text. You should also mention the errors associated with fine-structure estimates - particularly in regions away from the open ocean.

Assuming the energy dissipation in the thermocline (depth range <120m), we follow the Gregg89 model, where the observations agree with the predictions sufficiently well to suggest that the simplest way to obtain average dissipation rates over large space and time scales is through $N^2/(N_0^2) < (S_10^4)/(S_G M^4) > (Gregg, 1989). This model is commonly applied in the mid-latitude thermocline as our observations. That is why we use this simple but accurate model. On the other hand, Water man et al. (20$
$R\omega = âŇŤ V_Z^2/((N^2 \zeta_Z^2)) The main term omitted for us is (\zeta_z) the relative local change in buoyancy frequency from background : \zeta_z = ((N^2 - N_ref^2))/(N_ref^2) For our data this value is 0.9. This value can be related with the level of stretching and squeezing of isopyc$
$165). Saying that the results agree "within uncertainty bounds" was an overstatement, so we have changed this in the new version.$
$24, 245 - 246, 376 - 379).$

Lines 195-200: This should be in methods Done (line 191-199)

Line 203: Spatial resolution - be careful what you mean by this as really each data point is an average over 1200m by 15 m box Yes, we agree. We must distinguish between

the theoretical resolution of the seismic data and that of the diapycnal mixing maps. For seismic data, the vertical resolution (i.e. the capability to discern between neighboring reflectors) is given by the Rayleigh criteria, whereas the horizontal resolution (i.e. the part of a reflector covered within half a wavelength of the seismic signal) corresponds to the first Fresnel zone. For our acquisition system, medium properties, and target depth, these are 2 m and 15 m respectively (it is explained in Sallares et al., 2016). However, this does not represent the resolution of the mixing map. In this case, we are calculating spectra and diapycnal mixing within windows of 1200x15 m, so this could be taken as the approximate resolution of the map. We have modified the text accordingly (line 197-199).

Lines 215: What scale are you computing shear over i.e. dZ? Also how to you quantify buoyancy frequencies, N? (Line 221) dz is 10 m. To calculate buoyancy frequency we use the expression below, where density is obtained from the XCTD data: $N=\sqrt{(-g/_0(\delta(z))/\delta_z)}$

Lines 319: Shear to strain ratios tell you about the frequency content of the internal wave field. You might well expect higher inertial content (i.e high shear to strain ratios) near the surface due to wind generation. As we describe below the level of stretching and squeezing of isopycnals by internal waves, is close to 1. Near the surface we would expect higher inertial content, but we consider the whole thermocline, where we can see a trending a robust regularity over the whole profile (figure rev1-1). This variation is consistent with the process already described in Sallares et al. (2016).

Lines 325: How do dissipation estimates compare for GM and Batchelor parts of the MCS spectra? This may tell you something about the role of IW in generating the turbulence/GM assumptions As it is shown in figure rev.1-1 and it is explained above, the general patterns in the diffusivity maps obtained with the GM (a) and Batchelor (b) parts of the spectra (including location of maximum and minimum values) are very similar. It appears to be a clear correspondence between the two diffusivity maps. However, the values obtained from the Batchelor part of the spectra are much higher

than those obtained from the GM part. To us, this indicates the stronger influence of instabilities, rather than IWs, on diffusivity.

Line 334 and 351: Confusing regards what you are trying to report regards role of internal waves here - please clarify. As we explained above, our main point is that we do not see a direct relationship between IWs and mixing. Mixing appear to concentrate where IWs become unstable and instabilities develop, leading to turbulence. We have tried to clarify this message in the new version of the text (line 248-250, 334-336, 383-386).

Please also note the supplement to this comment:
https://www.ocean-sci-discuss.net/os-2017-72/os-2017-72-AC1-supplement.pdf

[Figure]

Figure rev.1-1. High-resolution $k_\rho(x,\,z)$ map overlapped with the HR-MCS image. Solid lines labelled H1 and H2, display acoustic reflectors located within relatively high- and low-dissipation areas (a) from internal waves Gregg89 model (b) from Batchelor59 model.

**Fig. 1.**

**Supplement:**

**General Comments:**

The paper presents some nice data acoustic data from the Alboran Sea, for which turbulent mixing estimates within the thermocline have been estimated at high spatial resolutions. mixing estimates show an interesting 'patchyness'. Some oceanographic data has also been used to compare mixing rate estimates. Overall the paper is well put together and the data well presented. However, I feel that the paper requires some further work on three counts:

We thank reviewer#1 for the constructive comments and review that has surely helped to improve the manuscript. We have taken all comments and suggestions into account as indicated in our point-by-point answers below.

1 - Clarification should be provided regards the physical mechanisms behind the mixing distribution observed. Messages regards the influence of internal waves seem confused.

We agree that the relationship between observed oceanographic features and mixing distribution was unclear. Our main message is that there is not a clear correspondence between the location of IWs (> 100 m horizontal scale) and mixing hotspots, but rather between mixing hotspots and the location of large-amplitude features in the transitional domain (30-100 m horizontal scale). Based on this analysis as well as on previous results presented in Sallares et al (2016), we interpret that these large-amplitude features are the expression of shear instabilities (e.g. KH-type billows). This means that there is not a direct relationship between IWs and mixing. It tends to concentrate where IWs become unstable and instabilities develop, leading to turbulence. We clarify this message in the new version of the text (line 19-21, line 339-341, line 383-386).

2 - The analysis of mixing-rates from oceanographic data, using fine-structure estimates (and assumptions therein) needs to be improved - see comments below

See answer below (comments line 165).

3 - I would like to see a comparison of mixing estimates form the internal wave and Batchelor regimes of the MCS spectra

We have estimated the $k_\rho(x,z)$ map for internal waves and Batchelor regimes (figure rev.1-1). The lower mixing values produced by IWs as compared to the Batchelor regime are clear.

[Figure]

Figure rev.1-1. High-resolution $k_\rho(x, z)$ map overlapped with the HR-MCS image. Solid lines labelled H1 and H2, display acoustic reflectors located within relatively high- and low-dissipation areas (a) from internal waves Gregg89 model (b) and from Batchelor59 model.

**Specific comments:**

Lines 35-37: Please clarify here - I think that both your references refer to internal-wave phenomena

Thanks for the comment, references have been corrected (line 38).

Lines 50-53: I am not sure what you are saying here.

We just want to describe the behavior of $\varepsilon$ in a conservative flow. We have modified the text to make this point clear (line 51-52).

Lines 65: I think that lowered microstructure profiles are generally the most robust source of turbulent measurements.

We agree, of course. Our point here was to note that these devices (VMP, microriders), provide measurements in just one dimension (either horizontal or vertical), but seismic data cover both dimensions at once. It is obviously with poorer resolution than microstructure profilers, but much better than that of conventional probe-based studies in the horizontal one (line 63-65)

Line 150: Do you have evidence for this? Could you compare acoustic reflection horizons to density horizons in the oceanographic data?

We do not have direct evidence for this particular profile because we cannot invert density with this data and we do not have the appropriate complementary oceanographic data.

However, in a previous study by our group with appropriate data, we showed that seismic reflectors do actually follow isopycnals (Biescas et al, 2014).

Line 165: If possible you should use integrated shear and or strain spectra to get estimates from CTD/ADCP data - perhaps you are limited by depth ranges? You are also missing some terms in for shear-to-strain ratio and inertial frequency e.g. see Waterman, S., K. L. Polzin, and A. C. Naveira-Garabato (2012), Internal waves and turbulence in the Antarctic Circumpolar Current, J. Phys. Oceanogr., 43, 259–282. You should at least quantify the omission of these terms and also explain how you decide on what you mean by 'uncertainty bounds' in several places in the text. You should also mention the errors associated with fine-structure estimates - particularly in regions away from the open ocean.

Assuming the energy dissipation in the thermocline (depth range <120m), we follow the Gregg89 model, where the observations agree with the predictions sufficiently well to suggest that the simplest way to obtain average dissipation rates over large space and time scales is through $N^2/N_0^2 < S_{10}^4/S_{GM}^4 >$ (Gregg, 1989). This model is commonly applied in the mid-latitude thermocline as our observations. That is why we use this simple but accurate model. On the other hand, Waterman et al. (2013) consider the relation:

$$R_\omega = \langle V_z^2 \rangle \Big/ (\overline{N}^2 \langle \zeta_Z^2 \rangle) \tag{1}$$

The main term omitted for us is $(\zeta_z)$ the relative local change in buoyancy frequency from background:

$$\zeta_z = (N^2 - N_{ref}^2)/N_{ref}^2 \tag{2}$$

For our data this value is ~0.9. This value can be related with the level of stretching and squeezing of isopycnals by internal waves, but as it is close to 1, the incidence is not relevant in our case (line 162-165).

Saying that the results agree "within uncertainty bounds" was an overstatement, so we have changed this in the new version. What we actually meant is that the global average and the values obtained with the XCTD are "within the range of values" obtained from the seismic data analysis. We have reworded the text accordingly (line 22-24, 245-246, 376-379).

Lines 195-200: This should be in methods

Done (line 191-199)

Line 203: Spatial resolution - be careful what you mean by this as really each data point is an average over 1200m by 15 m box

Yes, we agree. We must distinguish between the theoretical resolution of the seismic data and that of the diapycnal mixing maps. For seismic data, the vertical resolution (i.e. the capability to discern between neighboring reflectors) is given by the Rayleigh criteria, whereas the horizontal resolution (i.e. the part of a reflector covered within half a wavelength of the seismic signal) corresponds to the first Fresnel zone. For our acquisition system, medium properties, and target depth, these are ~2 m and ~15 m respectively (it is explained in Sallares et al., 2016). However, this does not represent the resolution of the mixing map. In this case, we are calculating spectra and diapycnal mixing within windows of 1200x15 m, so this could be taken as the approximate resolution of the map. We have modified the text accordingly (line 197-199).

Lines 215: What scale are you computing shear over i.e. dZ? Also how to you quantify buoyancy frequencies, N?

(Line 221) dz is 10 m. To calculate buoyancy frequency we use the expression below, where density is obtained from the XCTD data:

$$N = \sqrt{-\frac{g}{\rho_0}\frac{\delta_{\rho}(z)}{\delta_z}} \tag{3}$$

Lines 319: Shear to strain ratios tell you about the frequency content of the internal wave field. You might well expect higher inertial content (i.e high shear to strain ratios) near the surface due to wind generation.

As we describe below the level of stretching and squeezing of isopycnals by internal waves, is close to 1. Near the surface we would expect higher inertial content, but we consider the whole thermocline, where we can see a trending a robust regularity over the whole profile (figure rev1-1). This variation is consistent with the process already described in Sallares et al. (2016).

Lines 325: How do dissipation estimates compare for GM and Batchelor parts of the MCS spectra? This may tell you something about the role of IW in generating the turbulence/GM assumptions

As it is shown in figure rev.1-1 and it is explained above, the general patterns in the diffusivity maps obtained with the GM (a) and Batchelor (b) parts of the spectra (including location of maximum and minimum values) are very similar. It appears to be a clear correspondence between the two diffusivity maps. However, the values obtained from the Batchelor part of the spectra are much higher than those obtained from the GM part. To us, this indicates the stronger influence of instabilities, rather than IWs, on diffusivity.

Line 334 and 351: Confusing regards what you are trying to report regards role of internal waves here - please clarify.

As we explained above, our main point is that we do not see a direct relationship between IWs and mixing. Mixing appear to concentrate where IWs become unstable and

instabilities develop, leading to turbulence. We have tried to clarify this message in the new version of the text (line 248-250, 334-336, 383-386).

---

## Author Comment (AC2) · 1 Nov 2017

First, we want to thank referee#2 for her/his effort. We found the comments and suggestions very useful, and we have tried to answer and/or follow all of them as indicated in our point-by-point answers below. To clarify our answers, we add a file (Review2-answer1.pdf) as a supplement, easily to read for you.

Thanks for the comment. We first want to make clear that the goal of the paper is not presenting the details of the data processing and spectral analysis, nor developing a new method to estimate mixing. The method used to produce the diapycnal diffusivity map from seismic data is not new; it is analogous to that presented in previous works

(i.e. Sheen et al, 2009; Holbrook et al, 2013). In addition, the seismic data and their spectra were recently processed, analyzed and interpreted in detail in another paper by our group (Sallares et al. 2016). In fact, many of the questions raised by referee#2 are addressed in this paper. We have modified the text to make it clearer in the new version of the manuscript (line 100-102, 137-141). We would like to emphasize that our goal and original contribution of the paper are (1) producing a diapycnal mixing map of higher resolution than any previously existing one and (2) applying it for the first time to shallow waters (thermocline), a critical area to study mixing processes. We then try to interpret the observed features based on the results but also on our previous work. To do this, we use data acquired in the Alboran basin with high-resolution multichannel seismic system, which were presented, processed, analyzed and interpreted by Sallares et al. (2016). The basic points of the method applied to produce the maps are explained in this manuscript, and the details can be found in the other two works mentioned above. We clarify this in the new version of the manuscript (lines 186-189 rewritten).

The size of the window to calculate the spectra and to estimate the mixing values is always 1200 m wide x 15 m high. The difference with previous similar works is that the windows overlap with each other; The center of the window moves only 30 m in horizontal, and 3 m in vertical in each step. By doing this, the transition is smooth because we incorporate few new data in each new analyzed window. We can see the effect in figure rev2-1. (a) Mixing map obtained following a "conventional" way (i.e. no overlapping windows). In this case we apply a step of dx=1200 m, dz=6 m between 1200 m wide x 6 m high neighboring windows. (b) Mixing map obtained using 1200 m wide x 15 m high overlapping windows and a dx=30 m, dz=3 m step. The distribution and k(x,z) values is equivalent to (a) but display smoother transitions, making the map look more "realistic". This type of representation is new, but as we stated above, the method to estimate k(x,z) based on the horizontal wavenumber spectra of seismic reflectors is not new. We clarify all this in the new version of the manuscript (lines 197-199, 272-273, 298-300).
We cut the long tracks in 1200 m-long segments so that they fit inside the windows. This does not affect the spectrum at the spatial range analyzed. As an example, we analyze in figure rev2-2 a 16 km-long reflector (H3). We first calculate the spectrum for the whole reflector and we then split it in 10 segments (1.6 km each), and calculate their individual spectra as well as the average. The average spectra is very similar to the complete one in terms of energy and slope at the scale of interest. The details on the procedure followed to calculate the spectra can be found in Sallares et al. (2016) (line 193-194).

As we already explained above: (1) 30x3m is not the grid cell, it is just the step applied to analyze a new "1200 m-long x 15 m-high" window. (2) The size of the windows (so that of the actual grid cells) is always 1200x15m. (3) The tracks longer than 1200 m are cut into smaller segments that fit inside the window. Concerning resolution, we must distinguish between the theoretical resolution of the seismic data and that of the diapycnal mixing maps. For seismic data, the vertical resolution (i.e. the capability to discern between neighboring reflectors) is given by the Rayleigh criteria, whereas the horizontal resolution (i.e. the part of a reflector covered within half a wavelength of the seismic signal) corresponds to the first Fresnel zone. As we explain in Sallares et al. (2016), for our acquisition system, medium properties, and target depth, these are 1-2 m and 12-15 m, respectively. However, this does not represent the resolution of the mixing map. In this case, we are calculating spectra and diapycnal mixing within windows of 1200x15 m, so this could be taken as the approximate resolution of the map (in fact resolution is higher thanks to the "sliding window" approach). In summary, we do not claim that we are resolving structures of 30x3m, but the clear, larger-scale yellowish patches of 1-3 km-wide x 10-20 m-thick that are clearly identified in the map (better explained now in lines 209-212).

As we explained above, our study builds on previous work concerning both method and data. The method to produce diapycnal mixing maps based on horizontal wavenumber spectra of seismic reflectors is described in Sheen et al. (2009) and Holbrook et al.
(2013). The data, including acquisition system, MCS data processing, reflector track-ing, S/N estimation, spectral analysis and statistical analysis of the obtained spectra, are presented in detail in Sallares et al. (2016) (in the main documents and supplemen-tary material). We do not think that it is necessary to repeat what is already explained and shown in these papers, but we could add part of it as supplementary material if the referee and editor think otherwise (e.g. figs. Rev2-3 or 2-4). In any case, we have introduced several changes in the text to clarify this (line 100-101, 186-189).

As we explained above, the original data, including the 68 reflectors and the criteria to select and track them, are shown and described in Sallares et al (2016). As you can see in figure rev2-3, they are rather homogeneously distributed throughout the analyzed area (30-110 m depth), so most of the 1200x15m analyzing windows contain reflectors and contribute to create the map. The few that do not have enough data to calculate the spectra are shown in white. We clarify this in lines 137-141.

At this sub-mesoscale (∼window size) we apply the Gregg89 approach (Gregg, 1989), which considers the Garret-Munk model (Garret and Munk, 1979). The observations agree with the model predictions sufficiently well to assume that it describes the link between internal waves and turbulence. The interpretation is that the model is close enough to reality to capture the principal interactions scaling the turbulent dissipation in the thermocline (line 162-165).

Several single-track spectra, as well as the combined spectra of all reflectors for two different seismic profiles including the one analyzed here, are presented in detail in Sallares et al (2016). Both the single and the combined spectra (fig rev2-4) consis-tently show analogous spectral slopes and slope breaks at the same horizontal scales. Additionally, the spectral slopes coincide with theoretical estimations for three different, well-known sub-regimes: the Garret-Munk model for internal waves at >100 m, Kelvin-Helmholtz instabilities at ∼100-30 m, and Batchelor model for turbulent regimes (< 30 m) (line 137-141, 150-152).

This issue is also addressed in Sallares et al (2016). As it can be observed in fig rev2-4, the combined spectra of the 68 spectra show clear slope changes consistent with theoretical estimates for the three sub-regimes referred to above at precise wavenumbers (∼100 m and ∼30 m, respectively). The bound between the IW and shear instability regimes coincides with $lN=2\pi\Delta V/N$, where N is the buoyancy frequency, and $\Delta V$ is the root mean square amplitude of the velocity fluctuation about the mean, which is also calculated within the targeted depth range (30-110 m) from ADCP data. The same spectral slopes and bounds are also obtained in the other seismic profile analyzed in Sallares et al (2016). This behavior also holds for most of the individual tracks. Note that otherwise we would not obtain such clear trends in the combined spectra (fig rev2-4) (137-141, 152-156). Our interpretation in Sallares et al. (2016) is that the energy cascade between internal waves and turbulence at the sheared thermocline presents a distinct transitional subrange, possibly governed by vortex sheet dynamics. We suggest that the transition starts with the inset of shear instability along the stratified thermocline, follows with the development and rollup of KH billows, and ends with their breaking, collapse and dissipation. The energy needed to maintain these spectra comes from internal waves generated by tidal forcing at the Gibraltar strait, which are in turn subjected to a constant shear between the Atlantic and Mediterranean waters. Even though our analysis is local, the fact that the individual spectra display systematically the same transitional subrange at about the same scales, strongly suggests that this chain of processes is occurring continuously and simultaneous over the whole surveyed area (lines 150-152).

We do not refer to all sub-mesoscale structures "in general", but just to the internal waves that affect this region. The variations in diapycnal diffusivity show no clear correspondence with internal waves, but rather with the shear instability-like features identified in the transitional range between 100-30m (figures 7-8). We have modified the text to clarify this (line 19-21, 248-250, 339-341).

Thanks for the recommendation. We do agree and, in fact, we already examined the

turbulent subrange to check if there was any correspondence between the features observed in this subrange (<30 m) and in the other two, and with the location of "mixing hotspots". We show two examples for H1 and H2 in figures rev2-5 and 2-6, respectively. It appears that it could be (fig rev2-5d), but the problem is that this subrange is too close to the resolution limit, especially in the vertical dimension of the analyzed structures, so data are rather noisy and it does not allow extracting meaningful conclusions.

Saying that the results agree "within uncertainty bounds" was an overstatement from our side. We have changed this in the new version. What we actually meant is that the global average and the values obtained with the XCTD are "within the range of values" obtained from the seismic data analysis (compare figs 4 and 5). We have reworded the text accordingly (line 22-24, 376-379).

Thanks for the comment. First, we agree that the relationship between IWs and over-turning is unclear and not directly justified by our results, so we have dropped this part from the text. Second, the relationship between shear instabilities and mixing hotspots comes from the analysis of various reflectors, not just H1 and H2. Here we show another reflector (H4) that show a similar pattern to H1. What we actually see is a correspondence between areas showing high diapycnal diffusivity and the location of the largest-amplitude features in the transitional domain, which we interpret to correspond to shear instabilities (possibly KH billows) based -also- on the results of Sallares et al. (2016). We have reworded lines 260-263 to clarify this.

The average k values presented in figure 4a is just a reference to compare with the range of values that we obtain from the seismic data. This way we confirm that our values are consistent with the ones inferred using more conventional oceanographic methods (same order of magnitude). But we fully agree that the main point of our results is the patchy nature and the range of variability (of over 4 orders of magnitude) in k. In this sense, we agree that the mean MCS/XCTD values shown in fig 4 were misleading so we have deleted them and we have incorporated instead a shadowed rectangle indicating the range of values obtained in the maps, which coincide with the

range of values obtained from the XCTD (new figure 4).

We agree that the hydrographic data are limited. However, these are the only "quasi-synoptical" data that we have, and we think that it is valuable to incorporate them in the discussion. The fact that both the average values for the whole column as well as the range of variability obtained from the XCTD compare well with those obtained from the –completely independent- seismic data is, in our opinion, a relevant result that is worth mentioning.

As it is explained in Sallares et al. (2016), an important step towards the calculation of the slope spectra is to suppress the random noise from the data and concentrate the analysis in the frequency bands where signal is clear. This can be efficiently done by: (1) estimating the signal-to-noise ratio (S/N) in the different frequency bands, and (2) selecting and applying a band-pass frequency filter that maximizes S/N. To estimate S/N we have applied a cross-correlation-based analysis that consists of the following steps:

i) Band-pass filtering the data;

ii) Calculate the cross-correlation (CC) between each seismic trace and all its neighbors within a distance equal to the length of the shortest reflectors used in the spectral analysis, dCC=1,250 m. This is first done in the upper part of the profile (30-120 m), hence the section that we consider to contain the signal.

iii) Calculate the maximum value of the CC within a time window corresponding to the mean separation between contiguous reflectors, tCC =10 ms, for each couple of traces (MaxSigij);

iv) Calculate the average value of MaxSigij for each seismic trace along the whole profile (AvMaxSigi);

v) Repeat steps ii) to iv) for the bottom part of the profile (120-240 m), which we consider to be noise, to obtain AvMaxNoisei;
[Figure]

vi) Calculate the ratio S/Ni=AvMaxSigi/AvMaxNoisei for each seismic trace;

vii) Calculate the average value of S/Ni for all the seismic traces: < S/Ni>=S/N;

viii) Repeat steps i) to vii) for the next frequency band.

No, we have not formally analyzed the statistical correlation between different signals. What we mean is that there is a visual correspondence between different features, as it happens, e.g., between high values of diapycnal mixing and large-amplitude features in the transitional domain. We have changed the word "correlation" by less confusing ones as "visual correspondence", or similar, in the new version of the manuscript (line 292, 336, 383).

We have modified the figure caption and text as suggested to briefly describe the procedure as follows: Figure 6 shows the average horizontal spectrum of the vertical displacement of tracked reflectors ($\Phi_\varsigma$x) scaled by the local buoyancy frequency at the reflector depth (N/N0) to eliminate stratification effects, and multiplied by $(2\pi k x)2$ to enhance slope variations (blackline). The reference lines are theoretical slopes of Garrett-Munk internal wave model [Garret and Munk, 1979] (red line), Kelvin-Helmholtz instabilities [Waite, 2011] (blue line), and Batchelor's model for turbulence [Batchelor, 1959] (green line).

Please also note the supplement to this comment:
https://www.ocean-sci-discuss.net/os-2017-72/os-2017-72-AC2-supplement.pdf

[Figure]

[Figure]

**Figure rev2-1.** $k_p(x, z)$ map obtained along the seismic profile. (a) without sliding window, using window size (1200 x 6 m) just getting one point for each window move, (b) applying sliding window, using window size (1200 x 15m) with step (dx=30, dz=3m). The trends and values are equivalent, but (b) looks more continuous and, to us, more realistic.

**Fig. 1.**

[Figure]

[Figure]

**Figure rev2-2** (a) Depth-converted high-resolution multichannel seismic profile (Here we show a new horizon H3). (b) Horizontal spectrum of the vertical displacement of reflector H3. (blue line) considering the whole reflector. (gray lines) spectrum from the reflector split in ten 1.6 km-long segments. (red line) average spectrum from the 10 segments. The average of the 10 segments and the whole reflector show the same trends in the scales of interest.

**Fig. 2.**

**Figure rev2-3.** Processed and depth-converted HR-MCS images along profile IMPULS-3, with the tracked reflectors used in the spectral analysis superimposed (blue lines). The depth range of the tracked reflectors is 30-100 m. The inset is a zoom over the area encompassed by the dashed rectangles (fig S5 in Sallares et al., 2016).

**Fig. 3.**

**Figure rev2-4**. Average horizontal spectrum of the vertical displacement, scaled by the local buoyancy, obtained for the 68 reflectors (solid line) and their corresponding 95% confidence interval (2σ) (shaded area). The reference lines are the theoretical slopes corresponding to the GM79 model for the internal wave subrange (red line), Kelvin-Helmholtz instabilities for the transitional/buoyancy subrange (blue line), and Batchelor59 model for turbulence (green line). The dashed line follows the original, unfiltered part of the spectra in the region affected by harmonic noise arising from repeated shooting. This is eliminated by applying a stop band of 0.027 to 0.021 Hz.

**Fig. 4.**

[Figure]

**Figure rev2-5**. (a) Diapycnal mixing obtained along H1 (see details of calculation in the text). (b) Signal filtered at wavelength ranges of the IW sub-range (*3000-100 m*), (c) the transitional subrange (*100-33 m*), (d) and the turbulence subrange (33-13 m). The dashed red line identifies the "breaking point" referred to in the text.

**Fig. 5.**

[Figure]

**Figure rev2-6**. (a) Diapycnal mixing obtained along H2 (see details of calculation in the text). (b) Signal filtered at wavelength ranges of the IW sub-range (*3000-100 m*), (c) the transitional subrange (*100-33 m*), (d) and the turbulence subrange (33-13 m). The dashed red line identifies the "breaking point" referred to in the text.

**Fig. 6.**

[Figure]

a

Depth (m)
Distance (km)

b

$\mathrm{Log}_{10}\,K_p\,(\mathrm{m^2\,s^{-1}})$

c

Filtered
Displacement (m)

d

Filtered
Displacement (m)

Length (km)

**Figure rev2-7**. (a) Location of H4 reflector. (b) Diapycnal mixing obtained along H4. (c) Signal filtered at wavelength ranges of the IW sub-range (*3000-100 m*), (d) the transitional subrange (*100-33 m*).

**Fig. 7.**

**Supplement:**

This manuscript presents seismic reflection data from the Alboran Sea and outlines a method for producing a map of diapycnal diffusivity across one profile. The major conclusion is that the profile shows patchy turbulence on the scale of a few kilometers horizontally and 10-15 meters vertically. Further, the authors observe greater mixing in areas of internal wave instability. Results are compared to estimates of turbulence made from XCTD and ADCP data as well as background reference models. Additional analyses of filtered slope spectra examine the relationship between diapycnal mixing and the assigned internal wave and transitional subranges.

First, we want to thank referee#2 for her/his effort. We found the comments and suggestions very useful, and we have tried to answer and/or follow all of them as indicated in our point-by-point answers below.

The introduction and background are clearly written and well presented. However, the manuscript takes on the substantial challenge of developing a new method and presenting scientific conclusions at once, and in a relatively short format. As a result, I think many issues addressing the methods, presented data, clarity of conclusions, uncertainties, and reach of results are insufficient.

Thanks for the comment. We first want to make clear that the goal of the paper is not presenting the details of the data processing and spectral analysis, nor developing a new method to estimate mixing. The method used to produce the diapycnal diffusivity map from seismic data is not new; it is analogous to that presented in previous works (i.e. Sheen et al, 2009; Holbrook et al, 2013). In addition, the seismic data and their spectra were recently processed, analyzed and interpreted in detail in another paper by our group (Sallares et al. 2016). In fact, many of the questions raised by referee#2 are addressed in this paper. We have modified the text to make it clearer in the new version of the manuscript (line 100-102, 137-141).

We would like to emphasize that our goal and original contribution of the paper are (1) producing a diapycnal mixing map of higher resolution than any previously existing one and (2) applying it for the first time to shallow waters (thermocline), a critical area to study mixing processes. We then try to interpret the observed features based on the results but also on our previous work. To do this, we use data acquired in the Alboran basin with high-resolution multichannel seismic system, which were presented, processed, analyzed and interpreted by Sallares et al. (2016). The basic points of the method applied to produce the maps are explained in this manuscript, and the details can be found in the

other two works mentioned above. We clarify this in the new version of the manuscript (lines 186-189 rewritten).

Major Concerns: Data handling and methods are insufficiently explained. The authors need to be clearer about the stated resolution. It is not accurate to apply a 1200x15 m grid to a 30x3 m grid and claim improved resolution. Many of the 30x3 m cells will not have tracks in them. In fact, at CMP spacing of 7.5 m, you can only have 4 or 5 traces represented (depending how you treat them) and few realistic and meaningful spectra can be taken at that scale.

The size of the window to calculate the spectra and to estimate the mixing values is always 1200 m wide x 15 m high. The difference with previous similar works is that the windows overlap with each other; The center of the window moves only 30 m in horizontal, and 3 m in vertical in each step. By doing this, the transition is smooth because we incorporate few new data in each new analyzed window. We can see the effect in figure rev2-1. (a) Mixing map obtained following a "conventional" way (i.e. no overlapping windows). In this case we apply a step of dx=1200 m, dz=6 m between 1200 m wide x 6 m high neighboring windows. (b) Mixing map obtained using 1200 m wide x 15 m high overlapping windows and a dx=30 m, dz=3 m step. The distribution and $k_\rho(x,z)$ values is equivalent to (a) but display smoother transitions, making the map look more "realistic". This type of representation is new, but as we stated above, the method to estimate $k_\rho(x,z)$ based on the horizontal wavenumber spectra of seismic reflectors is not new. We clarify all this in the new version of the manuscript (lines 197-199, 272-273, 298-300).

[Figure]

**Figure rev2-1.** $k_\rho(x, z)$ map obtained along the seismic profile. (a) without sliding window, using window size (1200 x 6 m) just getting one point for each window move, (b) applying sliding

window, using window size (1200 x 15m) with step (dx=30, dz=3m). The trends and values are equivalent, but (b) looks more continuous and, to us, more realistic.

The authors need to explain what happens when tracks are longer than the 1200 m box.

We cut the long tracks in 1200 m-long segments so that they fit inside the windows. This does not affect the spectrum at the spatial range analyzed. As an example, we analyze in figure rev2-2 a 16 km-long reflector (H3). We first calculate the spectrum for the whole reflector and we then split it in 10 segments (1.6 km each), and calculate their individual spectra as well as the average. The average spectra is very similar to the complete one in terms of energy and slope at the scale of interest. The details on the procedure followed to calculate the spectra can be found in Sallares et al. (2016) (line 193-194).

[Figure]

**Figure rev2-2** (a) Depth-converted high-resolution multichannel seismic profile (Here we show a new horizon H3). (b) Horizontal spectrum of the vertical displacement of reflector H3. (blue line) considering the whole reflector. (gray lines) spectrum from the reflector split in ten 1.6 km-long

segments. (red line) average spectrum from the 10 segments. The average of the 10 segments and the whole reflector show the same trends in the scales of interest.

They state the tracks are 1.5-21 km long, so no tracks would fit inside the 1200 m grid length and position vertically is also unaddressed. Each track would be included in hundreds of 30x3 m grid cells which seriously undermine any claims at resolving patchy turbulence at their stated resolution.

As we already explained above: (1) 30x3m is not the grid cell, it is just the step applied to analyze a new "1200 m-long x 15 m-high" window. (2) The size of the windows (so that of the actual grid cells) is always 1200x15m. (3) The tracks longer than 1200 m are cut into smaller segments that fit inside the window.

Concerning resolution, we must distinguish between the theoretical resolution of the seismic data and that of the diapycnal mixing maps. For seismic data, the vertical resolution (i.e. the capability to discern between neighboring reflectors) is given by the Rayleigh criteria, whereas the horizontal resolution (i.e. the part of a reflector covered within half a wavelength of the seismic signal) corresponds to the first Fresnel zone. As we explain in Sallares et al. (2016), for our acquisition system, medium properties, and target depth, these are 1-2 m and 12-15 m, respectively. However, this does not represent the resolution of the mixing map. In this case, we are calculating spectra and diapycnal mixing within windows of 1200x15 m, so this could be taken as the approximate resolution of the map (in fact resolution is higher thanks to the "sliding window" approach). In summary, we do not claim that we are resolving structures of 30x3m, but the clear, larger-scale yellowish patches of 1-3 km-wide x 10-20 m-thick that are clearly identified in the map (better explained now in lines 209-212).

The authors need to show more of the data that support their methods and conclusions, particularly the reflector tracks and many more spectra.

As we explained above, our study builds on previous work concerning both method and data. The method to produce diapycnal mixing maps based on horizontal wavenumber spectra of seismic reflectors is described in Sheen et al. (2009) and Holbrook et al. (2013). The data, including acquisition system, MCS data processing, reflector tracking, S/N estimation, spectral analysis and statistical analysis of the obtained spectra, are presented in detail in Sallares et al. (2016) (in the main documents and supplementary material). We do not think that it is necessary to repeat what is already explained and shown in these papers, but we could add part of it as supplementary material if the referee and editor think otherwise (e.g. figs. Rev2-3 or 2-4). In any case, we have introduced several changes in the text to clarify this (line 100-101, 186-189).

To establish this as both a methods paper and support their science conclusion, these data must be shown and clear. First, show the 68 reflector tracks and discuss their distribution, including why application of a k value obtained on the large scale can be applied to the small scale and how to handle regions lacking tracked reflectors.

As we explained above, the original data, including the 68 reflectors and the criteria to select and track them, are shown and described in Sallares et al (2016). As you can see in figure rev2-3, they are rather homogeneously distributed throughout the analyzed area (30-110 m depth), so most of the 1200x15m analyzing windows contain reflectors and contribute to create the map. The few that do not have enough data to calculate the spectra are shown in white. We clarify this in lines 137-141.

[Figure]

**Figure rev2-3.** Processed and depth-converted HR-MCS images along profile IMPULS-3, with the tracked reflectors used in the spectral analysis superimposed (blue lines). The depth range of the tracked reflectors is 30-100 m. The inset is a zoom over the area encompassed by the dashed rectangles (fig S5 in Sallares et al., 2016).

At this sub-mesoscale (~window size) we apply the Gregg89 approach (Gregg, 1989), which considers the Garret-Munk model (Garret and Munk, 1979). The observations agree with the model predictions sufficiently well to assume that it describes the link between internal waves and turbulence. The interpretation is that the model is close enough to reality to capture the principal interactions scaling the turbulent dissipation in the thermocline (line 162-165).

Second, slope spectra rely on the aggregate data of many tracks to have statistically characteristic behaviors (Klymak and Moum, 2007 part II). All of what we are shown are single-track spectra.

Several single-track spectra, as well as the combined spectra of all reflectors for two different seismic profiles including the one analyzed here, are presented in detail in Sallares et al (2016). Both the single and the combined spectra (fig rev2-4) consistently show analogous spectral slopes and slope breaks at the same horizontal scales. Additionally, the spectral slopes coincide with theoretical estimations for three different, well-known sub-regimes: the Garret-Munk model for internal waves at >100 m, Kelvin-

Helmholtz instabilities at ~100-30 m, and Batchelor model for turbulent regimes (< 30 m) (line 137-141, 150-152).

[Figure]

**Figure rev2-4**. Average horizontal spectrum of the vertical displacement, scaled by the local buoyancy, obtained for the 68 reflectors (solid line) and their corresponding 95% confidence interval ($2\sigma$) (shaded area). The reference lines are the theoretical slopes corresponding to the GM79 model for the internal wave subrange (red line), Kelvin-Helmholtz instabilities for the transitional/buoyancy subrange (blue line), and Batchelor59 model for turbulence (green line). The dashed line follows the original, unfiltered part of the spectra in the region affected by harmonic noise arising from repeated shooting. This is eliminated by applying a stop band of 0.027 to 0.021 Hz.

Third, more justification of the horizontal wavenumber bounds for the sub-regimes (IW, instability/transition, and turbulent) are needed to illustrate these are accurate bounds for sub-regimes all over the 2D line. The authors need to be clearer about the role of basic oceanographic features and the expression of turbulent structures.

This issue is also addressed in Sallares et al (2016). As it can be observed in fig rev2-4, the combined spectra of the 68 spectra show clear slope changes consistent with theoretical estimates for the three sub-regimes referred to above at precise wavenumbers (~100 m and ~30 m, respectively). The bound between the IW and shear instability regimes coincides with $l_N = 2\pi\Delta V/N$, where N is the buoyancy frequency, and $\Delta V$ is the root mean square amplitude of the velocity fluctuation about the mean, which is also calculated within the targeted depth range (30-110 m) from ADCP data. The same spectral slopes and bounds are also obtained in the other seismic profile analyzed in Sallares et al (2016). This behavior also holds for most of the individual tracks. Note that otherwise we would not obtain such clear trends in the combined spectra (fig rev2-4) (137-141, 152-156).

Our interpretation in Sallares et al. (2016) is that the energy cascade between internal waves and turbulence at the sheared thermocline presents a distinct transitional subrange, possibly governed by vortex sheet dynamics. We suggest that the transition starts with the inset of shear instability along the stratified thermocline, follows with the development and rollup of KH billows, and ends with their breaking, collapse and dissipation. The energy needed to maintain these spectra comes from internal waves generated by tidal forcing at the Gibraltar strait, which are in turn subjected to a constant shear between the Atlantic and Mediterranean waters. Even though our analysis is local, the fact that the individual spectra display systematically the same transitional subrange at about the same scales, strongly suggests that this chain of processes is occurring continuously and simultaneous over the whole surveyed area (lines 150-152).

Lines 245-246 state "there is no clear visual correspondence between the k anomalies and the most obvious of the imaged oceanographic features such as IWs" which seems to be against the main thrust of the paper that sub-mesoscale features can be examined through their turbulent expressions, particularly lines 19-21 in the abstract as well as a few points in section 4.

We do not refer to all sub-mesoscale structures "in general", but just to the internal waves that affect this region. The variations in diapycnal diffusivity show no clear correspondence with internal waves, but rather with the shear instability-like features identified in the transitional range between 100-30m (figures 7-8). We have modified the text to clarify this (line 19-21, 248-250, 339-341).

The manuscript thesis is about ocean mixing, as such, the turbulent subrange should be examined for tracks H1 and H2. Perhaps there are corresponding traits between the IW subrange and turbulent subrange, or the transitional subrange and turbulent subrange that may clarify the expression of turbulence by mesoscale and sub-mesoscale oceanographic features. Uncertainties are problematic and under addressed.

Thanks for the recommendation. We do agree and, in fact, we already examined the turbulent subrange to check if there was any correspondence between the features observed in this subrange (<30 m) and in the other two, and with the location of "mixing hotspots". We show two examples for H1 and H2 in figures rev2-5 and 2-6, respectively. It appears that it could be (fig rev2-5d), but the problem is that this subrange is too close to the resolution limit, especially in the vertical dimension of the analyzed structures, so data are rather noisy and it does not allow extracting meaningful conclusions.

[Figure]

**Figure rev2-5**. (a) Diapycnal mixing obtained along H1 (see details of calculation in the text). (b) Signal filtered at wavelength ranges of the IW sub-range (*3000-100 m*), (c) the transitional subrange (*100-33 m*), (d) and the turbulence subrange (33-13 m). The dashed red line identifies the "breaking point" referred to in the text.

[Figure]

**Figure rev2-6**. (a) Diapycnal mixing obtained along H2 (see details of calculation in the text). (b) Signal filtered at wavelength ranges of the IW sub-range (*3000-100 m*), (c) the transitional subrange (*100-33 m*), (d) and the turbulence subrange (33-13 m). The dashed red line identifies the "breaking point" referred to in the text.

In the abstract (line 23) and conclusion (line 372) results are stated to be within uncertainty bounds but nowhere in the data do the authors show or discuss any uncertainty assessments. In section 3.2 uncertainty is briefly mentioned, but again, is simply stated that values are within uncertainty bounds. If the conclusions are supported within an uncertainty range, please show and elaborate.

Saying that the results agree "within uncertainty bounds" was an overstatement from our side. We have changed this in the new version. What we actually meant is that the global average and the values obtained with the XCTD are "within the range of values" obtained from the seismic data analysis (compare figs 4 and 5). We have reworded the text accordingly (line 22-24, 376-379).

The major conclusions for the filtered spectra analysis are under supported. Lines 378-381 deliver major conclusions about the relationship between IWs and overturning as well as shear instabilities and mixing hotspots. From the data presented, it appears these conclusions are drawn from the analysis of 2 tracked seismic reflections, H1 and H2. This overstates what is observed in the data, particularly when those two tracks were chosen as end-member individuals chosen for their position in "anomalously high (H1) and low (H2) mixing patches" (lines 264-265).

Thanks for the comment. First, we agree that the relationship between IWs and overturning is unclear and not directly justified by our results, so we have dropped this part from the text. Second, the relationship between shear instabilities and mixing hotspots comes from the analysis of various reflectors, not just H1 and H2. Here we show another reflector (H4) that show a similar pattern to H1. What we actually see is a correspondence between areas showing high diapycnal diffusivity and the location of the largest-amplitude features in the transitional domain, which we interpret to correspond to shear instabilities (possibly KH billows) based -also- on the results of Sallares et al. (2016). We have reworded lines 260-263 to clarify this.

[Figure]

**Figure rev2-7**. (a) Location of H4 reflector. (b) Diapycnal mixing obtained along H4. (c) Signal filtered at wavelength ranges of the IW sub-range (*3000-100 m*), (d) the transitional subrange (*100-33 m*).

General Comments: The manuscript thesis states that authors produce a map of diapycnal mixing that show patchy nature. However, they often refer to an average k as a benchmark to compare to conventional methods. The authors need to discuss why they would average the entire map when the main thrust of the paper is that it is heterogeneous.

The average $k_p$ values presented in figure 4a is just a reference to compare with the range of values that we obtain from the seismic data. This way we confirm that our values are consistent with the ones inferred using more conventional oceanographic methods (same order of magnitude). But we fully agree that the main point of our results is the patchy nature and the range of variability (of over 4 orders of magnitude) in $k_p$. In this sense, we agree that the mean MCS/XCTD values shown in fig 4 were misleading so we have deleted them and we have incorporated instead a shadowed rectangle indicating the range of values obtained in the maps, which coincide with the range of values obtained from the XCTD (new figure 4).

Additionally, the authors need to justify why just 1 XCTD and 1 ADCP data set would accurately reflect the average for their produced mixing map. For example, even if the XCTD was collected concurrently, what would the implication be if it was dropped at 8 km where k is high, or at 22.5 km where k is low?

We agree that the hydrographic data are limited. However, these are the only "quasi-synoptical" data that we have, and we think that it is valuable to incorporate them in the discussion. The fact that both the average values for the whole column as well as the range of variability obtained from the XCTD compare well with those obtained from the – completely independent- seismic data is, in our opinion, a relevant result that is worth mentioning.

Please explain how signal-to-noise is calculated. A signal-to-noise ratio of higher than 8 with 6-fold data is surprising.

As it is explained in Sallares et al. (2016), an important step towards the calculation of the slope spectra is to suppress the random noise from the data and concentrate the analysis in the frequency bands where signal is clear. This can be efficiently done by: (1) estimating the signal-to-noise ratio (S/N) in the different frequency bands, and (2) selecting and applying a band-pass frequency filter that maximizes S/N. To estimate S/N we have applied a cross-correlation-based analysis that consists of the following steps:

  i)       Band-pass filtering the data;

  ii)      Calculate the cross-correlation (CC) between each seismic trace and all its

neighbors within a distance equal to the length of the shortest reflectors used in the spectral analysis, $d_{CC}$=1,250 m. This is first done in the upper part of the profile (30-120 m), hence the section that we consider to contain the signal.

iii) Calculate the maximum value of the CC within a time window corresponding to the mean separation between contiguous reflectors, $t_{CC}$ =10 ms, for each couple of traces (*MaxSigij*);

iv) Calculate the average value of *MaxSigij* for each seismic trace along the whole profile (*AvMaxSigi*);

v) Repeat steps ii) to iv) for the bottom part of the profile (120-240 m), which we consider to be noise, to obtain *AvMaxNoisei*;

vi) Calculate the ratio *S/Ni=AvMaxSigi/AvMaxNoisei* for each seismic trace;

vii) Calculate the average value of *S/Ni* for all the seismic traces: < *S/Ni*>=S/N;

viii) Repeat steps i) to vii) for the next frequency band.

Much of section 3 (first paragraph) should be expanded and put into section 2

Ok, we have done this.

Please explain why tracks H1 and H2 were analyzed with 1 km windows when the turbulent maps were analyzed with 1.2 km windows. The difference in these windows change the wavenumber range of the IW spectra and might aid some of the confusion around the handling of internal waves in the manuscript.

It was a typo. The window size for H1 and H2 is the same as in the map (1.2 km-wide) (line 272, 299).

 Further, why limit the analysis to ~1 km? If they exist, larger IWs should carry even more energy and may be important.

At the spatial scale where we focus the analysis, the variation induced at longest wavelength does not affect directly the transitional subrange, as we confirm in figure rev2-2

At many points the authors state there are "no clear correlation" or similar language between filtered spectra (e.g. lines 287, 329-331). Were correlations and statistics taken for each of the 68 tracked reflectors to support, or not, a relationship between the filtered spectra? If so, this should be a major point of the paper and have supporting figures.

No, we have not formally analyzed the statistical correlation between different signals. What we mean is that there is a visual correspondence between different features, as it happens, e.g., between high values of diapycnal mixing and large-amplitude features in

the transitional domain. We have changed the word "correlation" by less confusing ones as "visual correspondence", or similar, in the new version of the manuscript (line 292, 336, 383).

In figure 6, please show the actual fit lines for these data. This would allow for a brief discussion in the text of how you calculate spectral energy levels beyond a reference to Sallares et al. 2016.

We have modified the figure caption and text as suggested to briefly describe the procedure as follows: Figure 6 shows the average horizontal spectrum of the vertical displacement of tracked reflectors ($\Phi\varsigma_x$) scaled by the local buoyancy frequency at the reflector depth ($N/N_0$) to eliminate stratification effects, and multiplied by ($2\pi k_x$)$^2$ to enhance slope variations (blackline). The reference lines are theoretical slopes of Garrett-Munk internal wave model [Garret and Munk, 1979] (red line), Kelvin-Helmholtz instabilities [Waite, 2011] (blue line), and Batchelor's model for turbulence [Batchelor, 1959] (green line).

**Specific comments:**

Line 85: Citation should be Holbrook et al., 2003.

Done

Line 175: Authors need to state and explain their choice for b the scale factor

It is the scale depth of the thermocline, where we focus the analysis and where we identified the internal waves.

Lines 329-330: Suggest rewording. The authors infer IW-induced mixing is not efficient enough to keep the overturning in this dataset, I do not think the data shown makes it clear, particularly on a global scale.

Reworded to reflect better the results.

Line 340: Suggest rewording. A smooth seafloor likely suggests a lesser role in the generation of hotspot mixing, if it is disregarded entirely, please explain.

Done

---

## Author Response (AR2)

Answer

**Topic Editor Decision: Reconsider after major revisions** (05 Jan 2018) by John M. Huthnance
Comments to the Author:
Dear Authors
Thank-you again for your revised version; you may now have seen the referee's comments on this revised version; for reference I have copied these comments below (sections A to I). The referee still wants to see your manuscript after further "major" revision (their term). Separately to me they have emphasised their concerns expressed in sections C, D, H below, and in respect of H also the question of the analysis for figure 8 also. I think the main message in all this is that your response to the referee's concerns should be embodied in your further-revised manuscript. The referee has emphasised that he does like the scientific significance of what you are writing about here, so please take all this as encouragement to clarify your manuscript as asked for.
Yours sincerely
John Huthnance

We thank the reviewer for the constructive comments and effort, which is helping much to improve the manuscript. We have taken all the comments and suggestions into account to rewrite a substantial part of the manuscript as indicated in our point-by-point answer below.

A) This manuscript presents a method of analysis aimed to quantify diapycnal diffusivity in the upper layers of the Alboran Sea using spectral methods. There are two major conclusions: first, it is possible to map the patchy nature of mixing in this data set. Second, there is a relationship between shear instabilities and mixing hotspots in the data but no correspondence between mixing and the location/amplitude of internal waves. The reported diffusivity levels generally match those found via using other methods: XCTD, ADCP, reference models.

As we mentioned above, we have modified the text in the new version of the m/s following the reviewer's comments. The first conclusion remains the same, but we have rewritten the part corresponding to the relationship between IWs, instabilities and mixing, which was confusing in the previous version. In summary, we do not mean that there is no relationship between IWs and mixing (we do not have direct evidence to discuss this and IWs are ubiquitous along the whole profile). What we say is that mixing appears to increase in areas with vigorous shear instabilities. Our interpretation is that shear instabilities are a mechanism enhancing energy transfer between IWs and turbulence (lines 19-22, 380-381, 441-447).

B) I think there is value in the results the authors attain. Products such as turbulence maps can teach us a lot about the oceanic interior and developing tools to do so using smaller seismic "high-res" hardware that can image the thermocline and shallow waters is a great step toward increasing the utility of seismic methods for oceanography. However, it is imperative to report findings as clearly and robustly as possible. I recognize that spectral methods are not the new innovation of this manuscript and am familiar with Sheen et al. 2009; Holbrook et al., 2013; and

Sallares et al., 2016; and the sliding analysis window for tracked seismic reflections is not a fundamentally new methodology either, as in Fortin et al., 2016. That said, I do consider this a new method due to the application to higher resolution data; it is not obvious that findings with large systems as in Sheen (2009) and Holbrook (2013) will equally apply to smaller systems. Additionally, using only tracked reflectors and a sliding window analysis has not been done, to my knowledge. Further, Sallares et al., 2016 is a short-format paper that shows only 1 spectra of the average of 117 tracks. Averages or sums of much data is required to get good spectra (Klymak & Moum 2007, a&b) so here when the authors start using smaller subsets of reflector tracks, it is necessary to show spectra sufficient to provide evidence their method is valid. The result of these adaptations is that the method used in this manuscript should be fully justified and show much supporting data.

We have made an effort to present our findings as clear and robust as we can, and interpreting the observed features based on these robust results. Concerning methodology, we essentially followed the indications of previously SO published works (e.g. Fortin et al. 2016; Holbrook et al., 2015) but adapting it to tracked reflectors within smaller widows. As suggested by the reviewer, and to clarify the main points of the procedure, we have added a more detailed, step-by-step description in the new version of the text:

The main steps of the procedure are the following ones: (1) Selecting a local window larger than the resolution of the data, but smaller that the entire seismic transect. The point is selecting the smallest possible window that allow calculating the reflector displacement spectra. We tested different window sizes and we found that the smallest ones that allow producing robust results are *1200 m* wide x *15 m* high. Results with larger windows are comparable in terms of amplitude and shape of the imaged features, but structures and boundaries are better defined with this window size (new Fig S1). (2) Calculating the spectra of all the reflectors inside the window (typically two to four for this window size) and average them. Examples of the results obtained for individual reflectors and average values obtained within different windows located in "high" and "low" mixing areas are shown in fig. 2rev 3 and new figure 6. Take the average value of the spectral amplitude between 13-30 m, which corresponds to the turbulent subrange (new Fig. S3). This range of scales can be resolved with the HR-MCS system used in this experiment, which has a theoretical lateral resolution of *8-17* m at the target depth (Sallares et al., 2016). (3) Applying Batchelor59 relationship (equation 7) to retrieve mixing rate based on the turbulent spectral amplitudes obtained within each window, and then Osborn80 (equation 1) to derive turbulent diffusivity from mixing rate. (4) As few tracked reflectors are included in the window, variances can affect the calculated diffusivities, then to eliminate this effect, we slide the window in small steps (30 m in horizontal and 3 in vertical direction), assigning the average value as explained above to every local window.

The analysis made for different window sizes and reflector lengths (new fig. S1) demonstrate the robustness of the results under different conditions and validate the procedure as a method to derive high-resolution turbulent diffusivity maps from MCS data.

A description of the procedure following steps 1-4 has been included in the new version of the m/s (lines 204-232).

We include in the new version of the manuscript a complementary analysis of reflectors recorded by our system. Besides, in figures 2rev. 2-3, we include much supporting data (windows and horizons analysis) that justify our findings in a robust way (new fig. 5-8, S1, lines 289-291; 301-305).

C) Many of the author's responses do not address the problems from the first review and the line numbers often point to incorrect sections of text. Specifically, here I am referring to the many times the line numbers refer to sections of text that are headers, blank lines, equations, or unchanged text [e.g. lines 100-102 (blank line in both manuscript versions and unchanged opening sentence); 162-165 (addition of sentence that doesn't add or address any concerns); 260-264 (only changed one word "anomalies" to "patches" which does not address the issues in the manuscript regarding k-rho values or distributions)]. The result is many changes that were difficult to track and, more importantly, often did not add to the clarity of the text or resolve the concerns posted in the prior review.

We apologize for having provided wrong number lines, we did not check it after the last edition of the previous version of the m/s. We have now checked that the comments and questions are addressed in the corresponding line numbers.

Lines 100-102 corresponded to lines 104-106. It refers to the method used to calculate the diapycnal mixing map.

Lines 162-165. We clarify and include the analysis of the 68 reflectors (figure 2rev1; new figure 2 and S3, lines 313-315) (Line 171-174), refers to the method used to obtain average dissipation rates.

Lines 260-264. It refers to the new window and horizon analysis that illustrates the relationship between shear instabilities and mixing hotspots (figure 2rev 3; new figure 5-8, lines 289-297).

[Figure]

*Figure. 2rev. 1. Depth-converted high-resolution multichannel seismic profile, with the tracked reflectors used in the spectral analysis superimposed (green lines).*

Major Concerns:
D) Turbulence levels in Figure 6 do not match how H1, H2, and H3 are described in the

manuscript or how they fit into the turbulence map of figures 3 and 5. In the text and due to its location in figures 2 and 5, horizon H1 is described at the high turbulence reflector, while H2 and H3 are low and moderate. However, figure 6 shows the energy in the turbulent subrange of H1 to be significantly lower than either H2 or H3. The authors need to explain what is happening here or else how can we be sure their map of turbulence is accurate when the "high turbulence" example shows less energy in the turbulent subrange.

We agree that the individual horizons chosen in the previous version were not the most appropriate to illustrate the relationship between spectral amplitude and turbulence level. This is because the analysis to calculate the spectra is done for individual windows containing several reflectors (see description of the method above, as well as in lines 204-212), so the relevant value for the map is the average of all reflectors within the window. To be consistent with this and to illustrate better our approach, we have changed the reflector-based analysis of the previous version of the m/s for a window-based one in the new version (figs 2rev 2 and 3, and new figs. 5 and 6). Now we can see several examples of reflectors within 1200 m wide x 15 m high windows located in "high" and "low" mixing regions (W1 to W6 in fig 2rev 1 and new fig 5), showing window average displacement slope spectral values in the turbulent subrange that are above and below the Batchelor59 model, respectively. These are the values taken to construct the turbulent diffusivity map.

[Figure]

*Figure 2rev. 2. High-resolution $k_\rho(x, z)$ map overlapped with the HR-MCS image. Squares indicate location of some of the 1200 m x 15 m windows analyzed. They have been selected as examples of high-dissipation (windows W1-W3) and low-dissipation (windows W4-W6) areas. The color code of the squares is the same as for reflector spectra in figure 2rev. 3, so that colors coincide with those of displacement spectra within the corresponding window.*

a.                                                                                     b.

[Figure]

*Figure 2rev. 3. Average horizontal spectrum of the vertical displacements of reflectors inside windows W1-W6 (see location and color code in figure 2rev. 2). (a) Spectra of individual reflectors in "high diffusivity" areas (thin dotted lines), average within windows W1 (red solid line), W2 (yellow solid line), and W3 (orange solid line), and average of the three "high diffusivity" windows (thick solid black line). (b) Spectrum of individual reflectors in "low diffusivity" areas (thin dotted lines), average within windows W4 (magenta solid line), W5 (green solid line), and W6 (blue solid line), and average of the three "low diffusivity" windows (thick solid black line). The reference lines are the theoretical slopes corresponding to the GM79 model for the internal wave subrange (brown dotted line), Kelvin-Helmholtz instabilities for the transitional subrange (dark blue dotted line), and Batchelor59 model for turbulence (dark green dotted line). Legend: Values of diapycnal diffusivity using spectral values at the turbulent subrange within each of the analyzed windows (same color code as for windows W1-W6).*

We have included this window-based analysis in the new version of the manuscript to illustrate the energy variation in the different areas and the relation between the mixing level and the oceanic processes (section 3.3, lines 301-318).

E) There are issues with the representation of resolution of the method. As stated in the abstract (line 25), the authors claim to resolve mixing with a lateral resolution on the order of 10 meters. The method applied, a rolling 1200 m x15 m analysis window, is attributing changes in spectral energy at the periphery of the window over half a kilometer away to the "high resolution" cell. It is unsurprising that, as stated in the conclusion (lines 397-398) the mixing hotspots appear to be 10-15 m vertically and 1-2 km laterally, a scale much closer to the real resolution of this treatment of the data. In their response the authors claim the method makes the turbulence maps appear more ' "realistic" ' and I agree that it does. The rolling window approach is not quite a smoothing function, but the authors should make certain the manuscript reflects the true resolution of the method.

We agree with the reviewer. The last sentence of the abstract was misleading so we have deleted it in the new version. It referred to the seismic system, not to the diffusivity map. As we explained in our previous response and was explained in the main text, the HR-MCS system has lateral resolution of O(10m) (~12-15 m in our case), but the mixing map was created with 1200 wide x 15 high sliding windows, so this is the approximate resolution of the map itself. We have emphasized this in the new version of the m/s (line 82-84 and 204-213).

F) Figures that are necessary, and have been produced by the authors, are missing from the manuscript.
- Figure rev2-3 shows the tracks of all the reflectors and detail of spatial coverage. This figure adds emphasis that a rolling window approach is potentially viable for this seismic line and a table or at least an average number of tracks per analysis window would make an excellent addition and help justify the handling of the data as was done in the manuscript.

Done, we include the figure in the main text (new Fig. 2), and the data requested in the manuscript (line 218-220).

- Figure rev2-2 shows much needed support for how the data was handled. The gray lines on this figure are hard to compare but show considerable variability. A re-working of this type of figure would give indication to just how certain we can be about analyses of a small number of reflector tracks broken into ~1-2 km segments.

Done, we include the figure 2rev. 3 in the main text (new Fig. 6).

- Figures rev2-5, rev2-6, rev2-7 include the turbulent subrange. This manuscript is fundamentally about turbulence and parts "d" from these figures should be included.

Done, we include the turbulence figures in the manuscript (new Fig. 7, and 8)

G) The relationship between internal waves and turbulence is still somewhat unclear. Lines 20-21 state "mixing tends to concentrate in areas where internal wave[s] become unstable and shear instabilities develop." This statement leads readers to think the data have a noted relationship between internal wave and turbulent structures. However, in the conclusion lines 406-407 say, "we found no clear correspondence between the location of the mixing patches and the location and amplitude of IWs" (also lines 255-256) then go on to discuss a relationship between shear instability and mixing. These two statements seem to be at odds with one another. The matter is further confused by introducing the relationship with shear instability. This reads as saying that (1) IWs and shear instability are related, (2) shear instability and turbulence are related, but (3) IWs and turbulence are not related. If both IWs and turbulence are related to shear instability, then it should follow that a relationship between IWs and turbulence would also exist.

Thanks for the comment. We agree that this deserves a clarification. As we mentioned above, we do not mean that there is no relationship between IWs and mixing. What we mean is that mixing increases in areas where IW instabilities develop. We interpret this as a sign that the development of shear instabilities is a mechanism that enhance energy transfer between IWs and turbulence. We hope this is now clearer in the new version (lines 19-22, 380-381, 441-447).

H) I am confused about the changes reported about analysis window length for figure 7a. Per previous review suggesting an analysis window of 1.2 km (to match the mapping analysis) the text was updated to say that figure 7a was done at 1.2 km in place of the first version using a 1 km window. However, this is no change in figure 7a from the previous version of the manuscript. I would expect a smoother profile of k-rho, particularly since the addition of 0.2 km is quite significant at the plotted scale and would encompass entire spikes and drops in turbulent signature. Further, figure rev2-2 shows great variability in non-overlapping 1.6 km segments, so there are significant differences at different points in the line and there should be differences when showing sliding 1.2 km segments as compared to 1 km segments.

Thank you. We rechecked the figures and now we include the ones with an analysis window length of 1.2 km. This way we get a smoother profile, and a clearer turbulence signature (new figure 7-8).

I) Minor Concerns
Please explain the scale factor "b" (line 182)

Done (Line 188).

Here we shown the marked-up manuscript version:

[revised manuscript text omitted]
 $\langle\varepsilon\rangle \approx 1.3\times10^{-8}\ Wkg^{-1}$, and an average diapycnal diffusivity $\langle k_\rho\rangle \approx \langle k_\rho\rangle \approx 10^{-3.0}\ m^2s^{-1}$ for the targeted depth range (Fig. 4a). The $k_\rho(z)$ profile obtained from the XCTD and ADCP is also shown in Fig. 4a, together with the global averages for overturning ($\langle k_\rho\rangle \approx 10^{-4}\ m^2 s^{-1}$) as well as the average pelagic diffusivity in the ocean ($\langle k_\rho\rangle \approx 10^{-5}\ m^2 s^{-1}$).

We obtain minimum values of the mixing rate at *50-55 m*, *68-73 m*, and *100-125 m*. The absolute minimum of $k_\rho = k_\rho = 10^{-5.2}\ m^2 s^{-1}$ is obtained at *~115 m*, whereas the maximum is of $10^{-2.1}\ m^2 s^{-1}$ at *~15 m*. This gives a range of variation of $10^{-3.1}\ m^2 s^{-1}$. Deeper than this,  mixing variability is smaller. The Turner angle and buoyancy frequency (Fig. 4b) indicate that the region is mostly stable with a slight tendency to double-diffusion ($Tu \approx 45°$).

It is worth noting that, at this specific location, the average vertical $\varepsilon(z)$ and $k_\rho(z)$ values are one order of magnitude higher that the global average ones. The higher values probably reflect the effect of overturning in the thermocline. While probe-based measurements are well suited to investigate mixing variability in the vertical dimension, they do not provide information on the variability in the horizontal dimension with a comparable level of detail. As explained above, to do this we have used estimations of $\varepsilon$ and $k_\rho k_\rho$ based on the HR-MCS data, but applying Batchelor59 model (Eq. 7) instead.

**3.2. High-resolution multichannel seismic-based $k_\rho(x, z)$ map**

The $k_\rho(x,z)$ map displayed in Fig. 3 has average values of $\langle\varepsilon\rangle \approx 6.5\times10^{-9}\ Wkg^{-1}$ and $\langle k_\rho\rangle \approx \langle k_\rho\rangle \approx 10^{-2.7}\ m^2 s^{-1}$. 
[revised manuscript text omitted]

860